# Irreversible furin cleavage site exposure renders immature tick-borne flaviviruses fully infectious

Jiří Holoubek[1,2,3], Jiří Salát[1,2,3], Milos Matkovic[4], Petr Bednář[1,2,3], Pavel Novotný[5,6], Martin Hradilek [5], Taťána Majerová [5], Ebba Rosendal[7], Luděk Eyer[1,2,3], Andrea Fořtová[1,2,3], Michaela Beránková[1,2,3], Lesley Bell-Sakyi[8], Anna K. Överby [7], Andrea Cavalli [4,9], Massimiliano Bonomi [10], Félix A. Rey [11] ✉ & Daniel Růžek [1,2,3] ✉

Flavivirus assembly is driven by the envelope glycoproteins pre-membrane (prM) and envelope (E) in the neutral pH environment of the endoplasmic reticulum. Newly budded, spiky particles are exported through the Golgi apparatus, where mildly acidic pH induces a major surface rearrangement. The glycoproteins reorganize into $(prM/E)_2$ complexes at the surface of smooth particles, with prM trapped at the E dimer interface, thereby exposing a furin cleavage site (FCS) for proteolytic maturation into infectious virions. Here, we show that in the absence of furin, immature tick-borne flavivirus particles—tick-borne encephalitis virus, Langat virus, and Louping ill virus—remain fully infectious and pathogenic in female BALB/c mice, in contrast to mosquito-borne flaviviruses such as Usutu, West Nile, and Zika viruses. We further show that the FCS in tick-borne viruses remains exposed at neutral pH, allowing furin at the surface of target cells to activate viral fusogenicity, while mosquito-borne counterparts require acidic re-exposure. Mutations increasing the dynamic behavior of the E dimer mimic the mosquito-borne phenotype, with retracted FCS at neutral pH and loss of infectivity. Our multidisciplinary approach—combining virological assays, targeted mutagenesis, structural modeling, and molecular dynamics simulations—highlights the role of E dimer dynamics in regulating flavivirus maturation and infectivity.

Flaviviruses—genus *Orthoflavivirus* (formerly genus *Flavivirus*), family *Flaviviridae*—are responsible for millions of human infections and numerous fatalities annually[1]. They constitute a diverse group of arboviruses transmitted by tick or mosquito vectors, which cause a spectrum of clinical syndromes in humans, including life-threatening encephalitides, haemorrhagic fevers, and birth defects[2]. Notable members include tick-borne encephalitis virus (TBEV; *Orthoflavivirus encephalitidis*)[3], West Nile virus (WNV; *Orthoflavivirus nilense*)[4],

[1]Department of Experimental Biology, Faculty of Science, Masaryk University, Brno, Czech Republic. [2]Laboratory of Emerging Viral Diseases, Veterinary Research Institute, Brno, Czech Republic. [3]Institute of Parasitology, Biology Centre of the Czech Academy of Sciences, Ceske, Budejovice, Czech Republic. [4]Institute for Research in Biomedicine, Università della Svizzera Italiana, Bellinzona, Switzerland. [5]Institute of Organic Chemistry and Biochemistry of the Czech Academy of Sciences, Prague, Czech Republic. [6]Department of Physical and Macromolecular Chemistry, Faculty of Science, Charles University, Prague, Czech Republic. [7]Department of Clinical Microbiology, Laboratory for Molecular Infection Medicine Sweden (MIMS), Umeå University, Umeå, Sweden. [8]Department of Infection Biology and Microbiomes, Institute of Infection, Ecological and Veterinary Sciences, University of Liverpool, Liverpool, United Kingdom. [9]Swiss Institute of Bioinformatics, Lausanne, Switzerland. [10]Institut Pasteur, Université Paris Cité, CNRS UMR 3528, Computational Structural Biology Unit, Paris, France. [11]Institut Pasteur, Université de Paris Cité, CNRS UMR 3569, Structural Virology Unit, Paris, France. ✉e-mail: felix.rey@pasteur.fr; ruzekd@paru.cas.cz

Japanese encephalitis virus (JEV; *Orthoflavivirus japonicum*)[5], Zika virus (ZIKV; *Orthoflavivirus zikaense*)[6], and dengue virus (DENV; *Orthoflavivirus denguei*)[7].

Mosquito-borne flaviviruses (MBFVs) most commonly cause human infections, but tick-borne flaviviruses (TBFVs) also constitute an emerging group that significantly impacts people across expansive geographic regions[8]. For instance, TBEV causes over 10,000 cases annually of tick-borne encephalitis (TBE) across Eurasia, predominantly affecting central, eastern, and northern Europe, and northeast Asia[9]. Recent detections of TBEV in the British Isles[10] and North Africa[11] indicate a broader geographic range for this virus.

Flaviviruses are small, enveloped viruses carrying an approximately 11-kilobase-long positive-sense single-stranded RNA genome (+ssRNA), including a single open reading frame that encodes a single, large precursor polyprotein. Co- and post-translational processing of the polyprotein by viral and cellular proteases[12] gives rise to seven non-structural and three structural proteins: capsid (C), pre-membrane (prM), and envelope (E)[13] which, together with the genomic RNA (gRNA), compose the viral particle. Protein prM binds to E co-translationally in the ER of the infected cell, acting as a chaperone for folding E into an active membrane-fusogenic protein. To be infectious, flavivirus particles require a maturation cleavage of prM by the host protease furin[14,15], yielding a globular, N-terminal half (termed pr) and M, the extended, viral membrane-anchored C-terminal half. The pr fragment remains associated to the mature particle as long the pH is acidic, blocking the fusion activity of E, but is expelled from the particles when they are exposed to the neutral pH external environment, activating the particle to infect other cells.

Flaviviruses enter target cells via receptor-mediated endocytosis, where the mildly acidic endosomal pH triggers the fusion of the viral envelope with the endosome in a process driven by protein E in the absence of pr. The viral gRNA/C nucleoprotein complex contained within the particle is thus released into the host cell cytoplasm[16,17], where the genome is translated by host ribosomes to generate the virus encoded proteins and initiate viral replication. The non-structural proteins induce invaginations of the endoplasmic reticulum (ER) membrane to form small membrane-bound replication organelles (RO[18]). On the other hand, the ER-membrane anchored prM/E heterodimers drive the budding of nascent viral particles that incorporate protein C in complex with the newly replicated viral genomes exiting the ROs[19,20]. These new particles are released within the ER lumen and feature 60 $(prM/E)_3$ trimers embedded in the viral membrane that interact laterally with icosahedral symmetry, projecting 60 trimeric spikes. Export to the extra-cellular milieu occurs through the Golgi apparatus, where the mildly acidic pH triggers a structural rearrangement of the virions, where the trimeric spikes dissociate and the individual protomers rearrange into 90 $(prM/E)_2$ dimers oriented tangentially to the cell membrane, forming a smooth surface layer[21]. Concomitantly, the acidic environment also allows the accessibility of a segment of prM containing the furin cleavage site (FCS), allowing particle maturation within the Golgi apparatus. Re-exposure to neutral pH upon release into the extracellular environment induces a final conformational change in E that expels pr from the particle[22], giving rise to fully mature virions coated by 90 $(M/E)_2$ dimers, assuming furin cleaves 100% of prM[15,23–25].

Studies with MBFVs have shown that the efficiency of furin cleavage is variable, depending both on the virus strain and on the host cell[26]. Partially mature virions were shown to retain infectivity, while fully immature particles were found to be non-infectious[15]. In the absence of furin cleavage, the conformational change of the virion from 60 $(prM/E)_3$ trimers to 90 $(prM/E)_2$ dimers was found to be reversible with pH for MBFVs[27,28], reforming a spiky arrangement in which the FCS retracts and is not accessible[27]. These results contrasted with the original experiments with TBEV, which had shown that once

exposed to acidic pH, the FCS remained accessible and furin could cleave when the particles were brought back to neutral pH[14]. Subsequent studies with the soluble ectodomain of protein E (sE) further showed that it had a different biochemical behavior with pH compared to the MBFV counterpart. TBEV sE behaves as a stable head-to-tail dimer[29] organized in the same way as E dimers on the mature particle surface[30–32], and which dissociates at mildly acidic pH[22]. In the case of MBFVs, sE behaves essentially as a monomer independently of pH, and requires a very high concentrations to form the sE dimers observed in the crystal structures[33,34]. Furthermore, for TBEV, recombinant pr binds to sE at acidic pH and blocks sE dimer dissociation by stabilizing a $(pr/sE)_2$ dimer[22], in a complex mimicking the pr bound E dimers of the smooth particles in the Golgi apparatus before reaching the cell surface, while for MBFVs recombinant pr does not stabilize an sE dimer[35]. The experimental structure of the immature TBEV spiky particle at neutral pH is available[36], but not at acidic pH nor at neutral pH after exposure to acidic pH, yet the behavior of the TBEV sE in complex with pr suggests that once formed, the herringbone pattern of E dimers of the immature particle remains when the particle is brought back to neutral pH and the FCS cannot retract, such that furin can cleave.

Here, we performed a comprehensive series of in vitro and in vivo experiments demonstrating that immature TBEV particles produced in furin knocked-out cells are fully infectious. We established this trait also for other TBFVs, including Langat virus (LGTV; *Orthoflavivirus langatense*) and louping ill virus (LIV; *Orthoflavivirus loupingi*). In contrast, and consistent with previous findings[37,38], immature particles from MBFVs—including Usutu virus (USUV; *Orthoflavivirus usutuense*), WNV, and ZIKV—displayed no infectivity or had significantly reduced infectivity. This newly identified divergence between TBFVs and MBFVs profoundly influences the pathogenic potential of TBEV and other TBFVs.

We also confirm that the proteolytic activity of furin is optimal at neutral pH[27], indicating that it is FCS accessibility that is impaired on the virion at neutral pH. In order to explore if FCS accessibility could be affected, we introduced two mutations just upstream the FCS in prM and one in E, at the E-dimer interface. The resulting mutant had a phenotype reminiscent of the mosquito-borne flaviviruses, with the immature particle losing infectivity and requiring acidic pH for cleavage by furin. Micro-second range molecular dynamics simulations of the triple mutant $(prM/E)_2$ and of the single mutant $sE_2$ dimer - using an AlphaFold prediction for the former and the experimental X-ray structure for the latter - showed increased dimer dynamics compared to wild type that correlates with the inhibited retraction of the FCS and infectiveness of the immature particles.

## Results

### Production and characterization of prM-TBEV

By infecting furin-deficient human LoVo cells, we produced immature prM-containing TBEV virions[39] (prM-TBEV) (Fig. 1a). These prM-TBEV and mature TBEV (m-TBEV, derived from BHK-21) were analysed by western blot. The m-TBEV sample was mostly cleaved, with a thin prM band indicating a low abundance of immature particles. In contrast, the prM-TBEV sample contained mostly uncleaved prM protein, indicating predominantly immature virions (Fig. 1b).

We quantified prM-TBEV in the supernatant using RT-qPCR and a plaque assay (Fig. 1c). Both analyses yielded similar titres: ~$10^5$ PFU ml$^{-1}$ in the plaque assay, and ~$10^6$ viral copies per ml by RT-qPCR. Surprisingly, this suggested that the prM-TBEV virions might be infectious.

We next tested prM-TBEV virus infection in three mammalian cell lines (PS, BHK-21, and Vero) and the vector tick cell line IRE/CTVM19. After infection at MOI = 0.1, viral infectivity was determined by plaque assay at 48 h post-infection (hpi) (mammalian cell lines) and 7 days post-infection (dpi) (IRE/CTVM19) (Fig. 1d). These results showed that prM-TBEV had infectivity similar to that of m-TBEV in all cell lines tested.

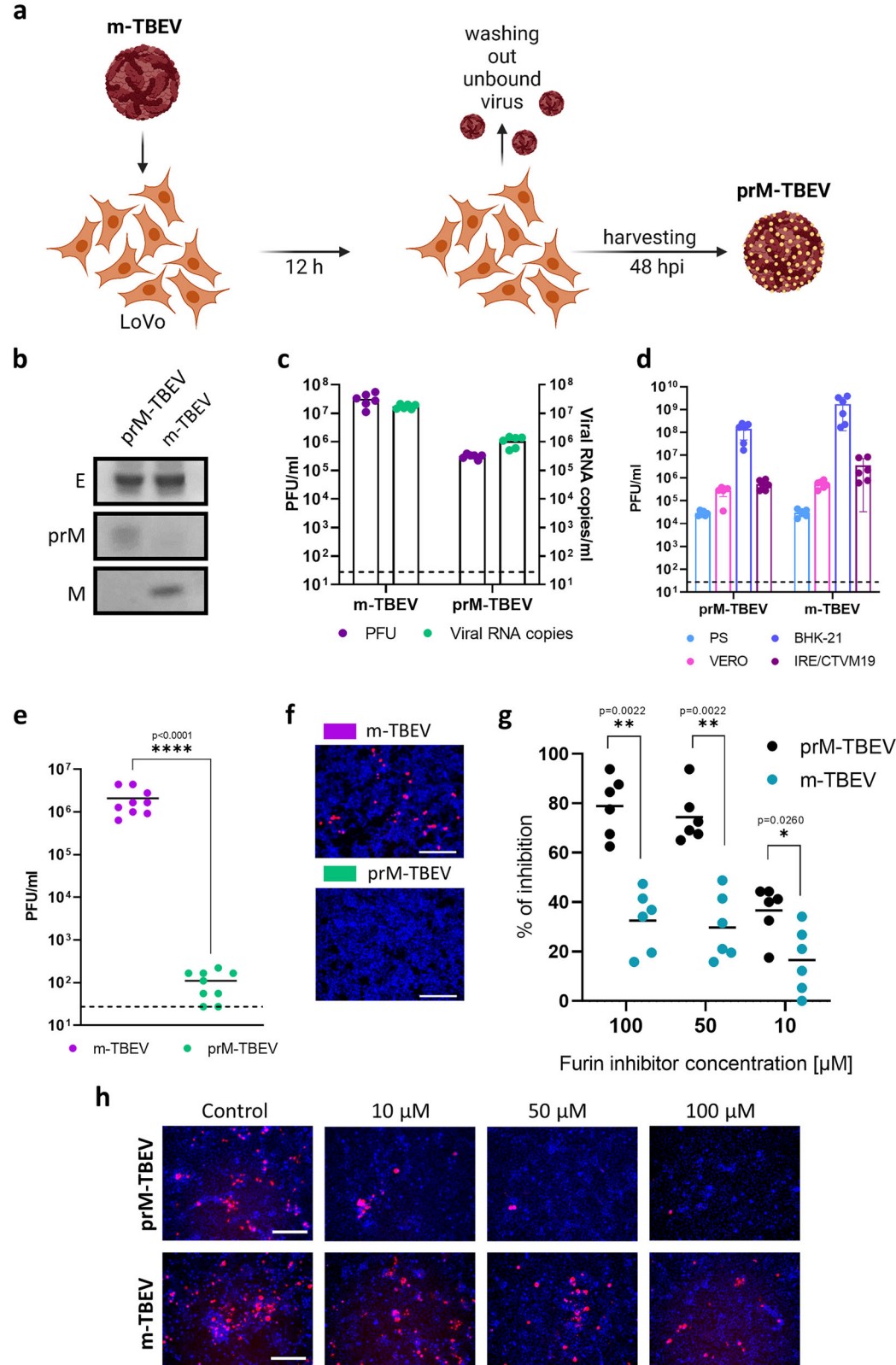

## Furin-mediated cleavage during virus entry

Furin cleavage is described mainly during virus maturation in Golgi apparatus (GA), but furin is also found in endosomes and on the cell surface[40]. To assess furin utilization during prM-TBEV entry, we infected LoVo cells with m-TBEV and prM-TBEV virus (MOI 0.1). Virus samples were harvested at 48 hpi, and the titres differed between groups by 4 $\log_{10}$ ($P < 0.0001$) (Fig. 1e).

Next, TBEV virus producing fluorescent mCherry protein was propagated in LoVo cells to obtain a prM-containing reporter variant (prM-mCherry-TBEV). LoVo cells infected with mCherry-TBEV exhibited a red fluorescent signal, which was absent in cells infected with prM-mCherry-TBEV (Fig. 1f). This suggested that the reduced prM-TBEV and prM-mCherry-TBEV infection was likely due to the absence of furin. We further used the furin inhibitor Decanoyl-RVKR-CMK to

**Fig. 1 | Production and characterization of immature prM-containing TBEV.**
**a** Schematic overview. LoVo cells (Δfurin) were infected with TBEV (MOI = 2). At 12 h post-infection (hpi), unbound virus was removed, the cells were washed with PBS, and fresh medium was added. At 48 hpi, prM-TBEV virus was harvested. **b** Western blot analysis of TBEV and prM-TBEV. Proteolytic processing of m-TBEV by furin yielded a completely cleaved prM protein. Without furin, the prM-TBEV sample contained a majority of immature prM-containing particles, as confirmed by the band corresponding to uncleaved prM protein. **c** Viral titres of m-TBEV and prM-TBEV determined by plaque assay, compared with viral RNA levels from quantitative PCR with reverse transcription (RT-qPCR). **d** Susceptibility of mammalian (PS, VERO, BHK-21) and tick (IRE/CTVM19) cell lines to prM-TBEV and m-TBEV. **e, f** Role of furin during host cell entry. LoVo cells were infected with TBEV and prM-TBEV (MOI = 0.1). The viral load in supernatant was determined by plaque assay, and

infectivity of the prM variant was visualized using mCherry_prM-TBEV and mCherry_TBEV. **g, h** To confirm that furin was necessary during entry steps of prM-TBEV infection, PS cells were pretreated with the furin inhibitor decanoyl-RVKR-CMK (100, 50, 25, and 0 μM). Cells were then infected with mCherry variants of the virus, as described above. At 48 hpi, infection was visualized, and the percentage of inhibition was determined by plaque reduction assay. Each image is representative of two separate experiments (*n* = 3). Dashed lines correspond to the detection limit of the plaque assay. Scale bars = 200 μm. Data from experiments presented in **c**−**e** are from biological duplicates or triplicates (*n* = 3) presented as mean values ± SD. The statistical significance was calculated using Mann-Whitney test for comparing two groups; ns, *P* > 0.05, *P* < 0.05, **P* < 0.01, ***P* < 0.001, ****P* < 0.0001. Schematic elements in panel a and selected annotation graphics were created in BioRender. Ruzek, D. (https://BioRender.com/4n4b27g).

ablate furin cleavage in a plaque reduction assay. Confluent PS cells were pretreated for 2 h with furin inhibitor (0, 10, 50, and 100 μM) and then infected with mCherry-TBEV and prM-mCherry-TBEV. Significant inhibition of prM-mCherry-TBEV by almost 80% was observed, especially at 50 and 100 μM (*P* < 0.001). In comparison, mCherry-TBEV was inhibited by only 35% (*P* < 0.01) (Fig. 1g, h). Together, these results confirmed the necessity of furin cleavage for maturation and demonstrated that this final maturation step can be efficiently performed during entry.

### Immature prM-flavivirus infectivity in TBFV and MBFV
We further investigated whether prM-containing immature particles of other TBFV and MBFV were infectious, as shown for prM-TBEV. Using LoVo cells, we produced prM samples for other flaviviruses transmitted by ticks (LGTV, LIV) and mosquitoes (USUV, WNV, ZIKV). Next, we performed a plaque assay with each prM flavivirus sample, to determine the numbers of infectious virions. All tick-borne prM variants showed high infectivity for mammalian cells, with titres of ~$10^4$ PFU ml$^{-1}$ (prM-LGTV), ~$10^5$ PFU ml$^{-1}$ (prM-TBEV), and ~$10^7$ PFU ml$^{-1}$ (prM- LIV). In contrast, MBFV prM showed low infectivity (prM-WNV ~$10^2$ PFU ml$^{-1}$) or no infectivity (prM-ZIKV and prM-USUV) (*P* < 0.0001) (Fig. 2a).

To confirm that MBFV could productively replicate in LoVo cells and produce immature non-infectious virions, we performed RT-qPCR and calculated the ratio of viral RNA copies to infectious plaque-forming units for immature and mature USUV and WNV (Fig. 2b, c), and TBEV (Fig. 2d). These results showed that the specific infectivities of prM-USUV and prM-WNV were at least 1,000-fold and 10,000-fold lower than those of m-USUV (*P* < 0.01) and m-WNV (*P* < 0.01), respectively. In contrast, the specific infectivities of prM-TBEV and m-TBEV differed by less than 10-fold (*P* < 0.01). Although TBFV and MBFV are closely related, their immature prM virions showed dramatically different infectivities, which could have further implications regarding their overall biology.

### pH-dependent furin cleavage in TBEV and USUV
To assess the pH dependence of furin cleavage, we performed in vitro cleavage experiments using recombinant human furin (r-furin). prM-TBEV infection of LoVo cells was examined at neutral (7.5) and acidic (5.5) pH, with or without r-furin. Under both pH conditions, LoVo cells were infected in the presence of r-furin (titres > $10^5$ PFU ml$^{-1}$), but not in its absence (*P* < 0.0001) (Fig. 2e). In addition, pH dependence was investigated using a western blot, which confirmed that furin cleavage is efficient throughout the pH range tested (Supplementary Fig. 2), implying that furin cleavage can occur during the entry steps. We next tested the infection of PS cells with prM-USUV, which occurred only at acidic pH, with or without r-furin (*P* < 0.0001). This indicated that endogenous furin from host cells could be utilized during prM-USUV infection at acidic pH (Fig. 2f,g). In comparison, prM-TBEV infected PS cells at both pH values, with or without r-furin (Fig. 2g). Together, these data showed that maturation cleavage of prM-TBEV virions occurs

independently of pH, whereas pH has important effects on maturation cleavage of prM-USUV. This highlights different pH-dependent dynamics of structural proteins, with irreversible or reversible structural changes observed for TBFV and MBFV, respectively (Fig. 2h).

### Pathogenicity of prM-TBEV and prM-WNV in a mouse model
In a mouse model, we examined the pathogenicity of prM-TBEV and prM-WNV, together with m-TBEV and m-WNV. For each virus variant, we established three groups of female BALB/c mice, which were inoculated subcutaneously (s.c.) with different doses: $10^1$, $10^2$, or $10^3$ viral genomic equivalents (gen. eq.). For 28 days, survival rates and clinical signs were monitored and scored. Doses of $10^2$ and $10^3$ gen. eq. of m-TBEV or m-WNV showed lethality rates of ~90%, as previously determined (Fig. 3 and Supplementary Fig. 3a, b). The highest doses of mTBEV and prM-TBEV caused fatal infections in all mice between days 8 and 12 (Supplementary Fig. 3a). Among mice infected with $10^2$ gen. eq., one mouse survived infection with m-TBEV (Fig. 3a). With the lowest infectious dose, 60% of m-TBEV-infected mice survived, compared to 40% of prM-TBEV-infected mice (Supplementary Fig. 3a). Overall, prM-TBEV and m-TBEV showed similar pathogenicity and reached similar viral titres in sera and brains collected 3 and 8 dpi, respectively (Fig. 3d, e).

In experiments with m-WNV, $10^3$ and $10^2$ gen. eq. resulted in lethal infection, whereas we observed 60% survival following infection with $10^1$ gen. eq. (Fig. 3f–h, and Supplementary Fig. 3b). Interestingly, all mice survived following infection with $10^1$ and $10^2$ gen. eq. of prM-WNV (*P* < 0.0001), without signs of neuroinfection, confirmed by plaque assay from the sera and brains (Fig. 3i,j). Only 20% of mice survived following infection with $10^3$ gen. eq. of prM-WNV (Supplementary Fig. 3b). Using calculations based on a ratio of PFU ml$^{-1}$ to viral RNA copies per ml and with an infection dose of $10^3$ gen. eq., inoculation of prM-WNV contained approximately 7 infectious virions per mouse. This caused fatal disease in 80% of mice, with a course of infection similar to that in mice infected with $10^1$ gen. eq. of m-WNV (~10 PFU/mouse). Overall, prM-WNV exhibited dramatically reduced infectivity in vivo.

### The proteolytic activity of furin is optimal at neutral pH
The proteolytic activity of furin and its pH dependence could be influenced by different amino acid compositions of the cleavage site and neighboring regions. The prM and E proteins of tick-borne and mosquito-borne flaviviruses share ~30–40% amino acid sequence identity (Supplementary Figs. 4, 5) and involve intricate interactions at the prM cleavage site and E protein interface. We hypothesized that polar/ionic interactions in and around the furin cleavage site may control its steric accessibility. To understand the intrinsic cleavability of the MBFV vs TBFV furin site sequences, we designed four internally quenched fluorogenic substrates based on prM sequences of two TBFVs (TBEV, LGTV) and two MBFVs (WNV, USUV), conjugated to Edans (donor for the FRET assay) and the quencher Dabcyl (Fig. 4a). For each substrate, Km values (Michaelis constant) were measured in

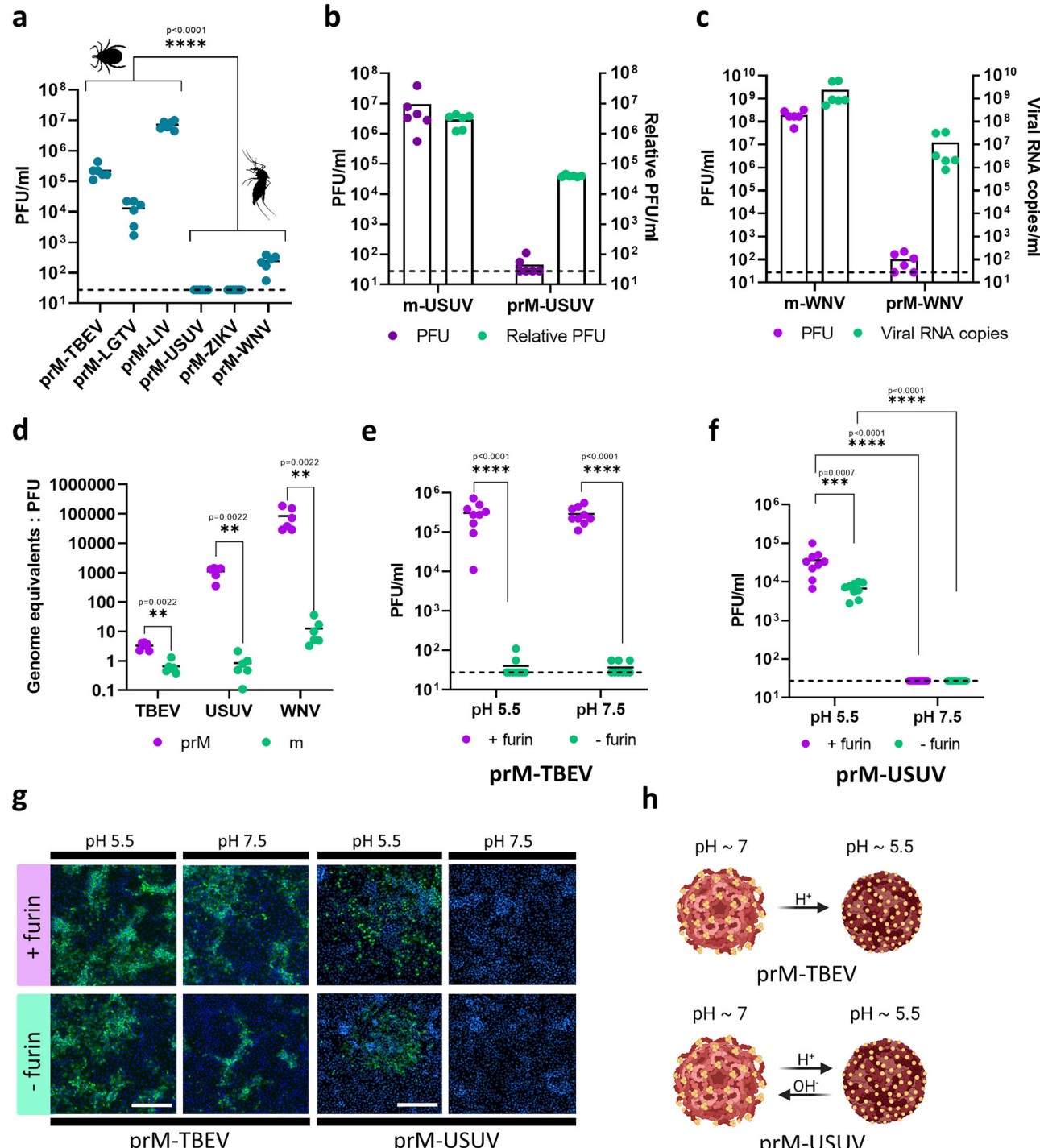

**Fig. 2 | Tick- and mosquito-borne flaviviruses and their sensitivity to pH. a** TBFV and MBFV were propagated in LoVo cells to produce prM variants. The infectivity of prM samples was determined 48 h post infection (hpi) by plaque assay. **b**, **c** To confirm that MBFV could replicate in LoVo cells, viral RNA levels were quantified by RT-qPCR, and the level of viral RNA copies per ml (rPFU ml⁻¹) was compared to PFU ml⁻¹. **d** These ratios showed dramatically decreased infectivity for prM-USUV and prM-WNV. **e** In vitro furin cleavage of prM-TBEV at neutral or acidic pH, followed by infection of LoVo cells (at neutral/acidic pH). The amount of infectious virions was determined 48 hpi by plaque assay. **f** In vitro cleavage of prM-USUV was performed at neutral or acidic pH, followed by infection of PS cells (at neutral/acidic pH). After 48 hpi, PFU mL⁻¹ was determined by plaque assay. **g** Immunofluorescence of prM-

TBEV and prM-USUV infection showed differing pH dependence of furin cleavage. Blue signal (DAPI) = cell nuclei; green signal = viral envelope protein. Scale bar = 200 μm. **h** Scheme of reversible/irreversible conformational changes induced by neutral and acidic pH. Each image is representative of at least two separate experiments (*n* = 3). Data were statistically analysed using the Mann-Whitney test. Data from experiments presented in **a–f** are from at least two separate biological experiments (*n* = 3) presented as mean values ± SD. The statistical significance was calculated using Mann-Whitney test for comparing two groups; ns, *P* > 0.05, **P* < 0.05, ***P* < 0.01, ****P* < 0.001, *****P* < 0.0001. Schematic elements in panel a and selected annotation graphics were created in BioRender. Ruzek, D. (https://BioRender.com/4n4b27g).

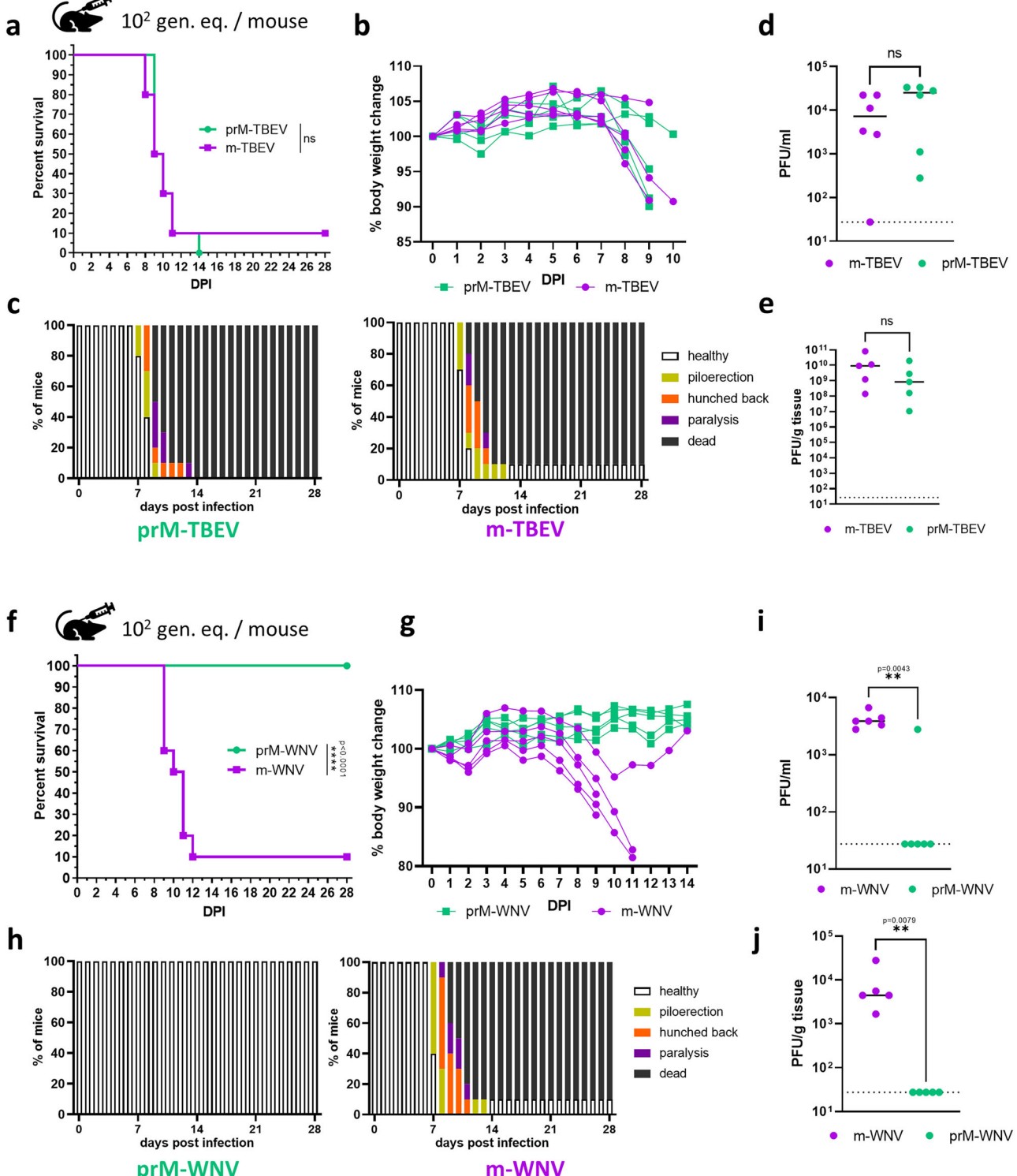

**Fig. 3 | Comparison of the pathogenicity of prM-TBEV/m-TBEV and prM-WNV/ m-WNV in a mouse model. a** Two groups of adult BALB/c mice (total n = 11 per group in two independent experiments) were subcutaneously infected with a dose of $10^2$ genome equivalents per mouse (gen. eq.) of prM-TBEV or m-TBEV, respectively. In the case of prM-TBEV, the infectious dose calculation corresponded to ~ 40 PFU for $10^2$ gen. eq. per mouse. Survival rates and clinical scores were monitored for 28 days. **b** Body weight changes were monitored and showed a similar pattern in all groups (6 mice per group, single experiment). **c** Clinical score was evaluated as follows: 1, no signs; 2, piloerection; 3, hunched back; 4, paralysis; and 5, death (total n = 11 mice per group, two experiments). **d** Viral titres in serum were evaluated 3 dpi. (6 mice per group, single experiment). **e** Mouse brains were harvested 8 dpi and viral titres were determined by plaque assay (5 mice per group, single

experiment). **f** As in the above experiment, two groups of mice were infected with either prM-WNV or m-WNV, and survival was monitored for 28 days (total n = 11 per group in two independent experiments). In the case of prM-WNV, the infectious dose calculation corresponds to ~ 0.7 PFU for $10^2$ gen. eq. per mouse. **g** Mouse body weight was monitored for 14 days (6 mice per group, single experiment). **h** Clinical signs of disease were assessed as described above (total n = 11 mice per group, two experiments). **i, j** Viral titres in serum (6 mice per group, single experiment) and brains (5 mice per group, single experiment) were measured by plaque assay 3 and 8 dpi, respectively. Survival rates were statistically evaluated using the log-rank Mantel-Cox test. Mouse and syringe icons were obtained from Microsoft PowerPoint's icon set.

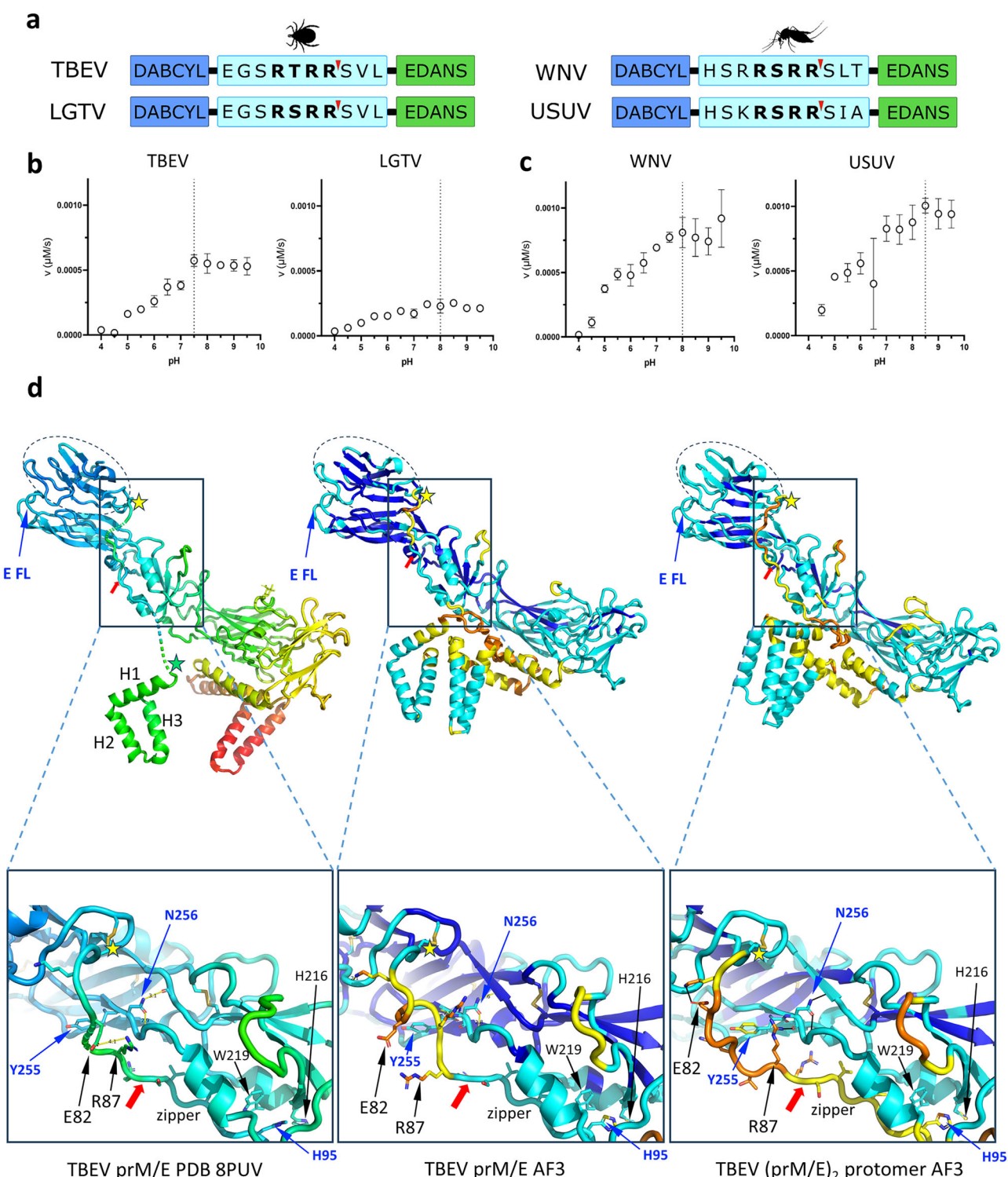

**a**

TBEV  DABCYL—E G S **R T R R**▸S V L—EDANS
LGTV  DABCYL—E G S **R S R R**▸S V L—EDANS

WNV  DABCYL—H S R **R S R R**▸S L T—EDANS
USUV  DABCYL—H S K **R S R R**▸S I A—EDANS

**b** TBEV / LGTV

**c** WNV / USUV

**d**

TBEV prM/E PDB 8PUV | TBEV prM/E AF3 | TBEV (prM/E)₂ protomer AF3

fluorescence assays, showing that MBFV-derived peptides were good substrates for r-furin (Km values: $1.6 \pm 0.33\,\mu M$ for WNV, and $2.1 \pm 0.35\,\mu M$ for USUV) (Supplementary Fig. 6). In contrast, the TBFV-derived peptides were poorer substrates (Km values: $4.3 \pm 0.68\,\mu M$ for TBEV, and $5.0 \pm 0.55\,\mu M$ for LGTV) (Supplementary Fig. 6). We next investigated the pH profile (4.0–9.5) of each substrate around its Km value. All tested substrates had similar and slightly alkaline optimal pH values, ranging from pH 7.5–8.5 (Fig. 4b, c), as observed previously with dengue virus[27]. Overall, MBFV-derived prM peptides were better substrates for furin than TBFV-derived substrates–in line with the presence of an additional basic residue (R or K) immediately preceding

the site (Supplementary Fig. 5). These data indicate that the direct interaction between furin and its cleavage site and flanking residues is not pH-dependent outside of the prM/E context, and their individual aa sequences do not cause the observed differences in maturation between TBFVs and MBFVs (Fig. 4a–c).

## prM/E zippering at neutral pH restricts furin accessibility to the cleavage site

The atomic model of the prM/E heterodimer derived from the recently reported cryo-EM structure of the neutral-pH immature TBEV spiky particle[36] showed traceable density for the FCS (prM

**Fig. 4 | Biochemical and structural characterization of the furin cleavage site in flaviviruses. a** Peptide substrates for furin activity assays were designed using the DABCYL-EDANS FRET pair. Each substrate includes five amino acids flanking both sides of the scissile bond to mimic the natural cleavage context. **b, c** The pH profile of each substrate was determined at a fixed concentration close to its Km at pH 7.0: HSRRSRRSLT (2.5 µM), HSKRSRRSIA (3 µM), EGSRSRRSVL (8 µM), and EGSRTRRSVL (5 µM). Reactions were performed in triplicate, and data are shown as mean ± SD. **d**, The prM/E heterodimer. The left panel shows the prM/E protomer from the experimental cryo-EM structure of an immature particle of TBEV at neutral pH (PDB 8PUV36). The two other panels show AlphaFold predictions, the middle panel is a prediction of the prM/E heterodimer, the right panel is the prediction of a (prM/E)2 dimer, but only one prM/E protomer is shown, for clarity (see Supplementary Fig. 4c). The limits of the linker region containing FCS are indicated by two stars (yellow and green) in the left panel. A dotted ribbon in the left panel (near the

green star) indicates a disordered region of the linker. The framed region is zoomed beneath each panel, slightly rotated for clarity. A red arrow points to the scissile bond, showing that it is located at the site where the zipper enters the groove underneath the E protein. The side chains of residues discussed in the text are shown as sticks, including Asn255 in E and the hydrogen bonds it makes drawn in dotted lines. The side chain of Tyr255 is also drawn, to show that the polypeptide chain of the linker, in between residues 82 and 87, interferes with contacts with Pro210 (in the second protomer, not shown) across the E dimer interface. The experimental structure is displayed as ribbons colored coded according to temperature (B) factors in the order lowest-blue < cyan < green < yellow < red-highest. The AF3 predictions are shown color-coded by plDDT (orange < 50, yellow > 50, cyan > 70, blue > 90). Schematic elements in panel **a** were created in BioRender. Ruzek, D. (https://BioRender.com/4n4b27g).

85-RTRR-88) and the region immediately downstream, hereafter termed the zipper, residues 89-SVLIPSHA-96, as it zippers along a groove formed between helices αA and αB of E (Fig. 4d and Supplementary Fig. 7). This organization shows that the zipper restricts accessibility of the FCS, as the scissile bond is in close interaction with the E protein at the groove entrance (Fig. 4d). The cryo-EM structures of the mature TBEV virion at neutral pH[41,42] also show the zipper (M residues 1-SVLIPSHA-8) inserted in the αA-αB groove in E and the M loop (a region that is disordered in prM in the immature particle, shown by dotted ribbons in Fig. 4d, left panel) ordered underneath the E dimer. These structures therefore predict that there must be unzippering at acidic pH to expose the scissile bond. In line with this conclusion, previous studies with TBEV M-E complexes detergent-solubilized from purified virions have shown that the interaction between E and M, which is essentially at the zipper/αA-αB groove since pr is absent, is pH sensitive, and is broken at acidic pH[43] due to the repulsion of His7 in M (or His95 in prM) and His216 in E (labeled in Fig. 4d, lower panels) (Supplementary Fig. 7), whose the side chains are close to each other within the groove[36,41,42]. It is also compatible with observations in dengue virus–which, like most of the flaviviruses, also has histidine at the equivalent of TBEV E His216 (Supplementary Fig. 4)-and which show that mutation of the equivalent of TBEV prM H95 – a strictly conserved histidine in all flaviviruses - rendered the virus maturation-deficient as prM was not cleaved by furin at physiological mildly acidic pH, and required a more stringent acid pH treatment for maturation. This same study showed that mutations elsewhere restored matura-tion at physiological pH, in line with certain flaviviruses (such as yellow fever or Spondweni viruses) not having histidine at the equivalent of H216 (Supplementary Fig. 4) but yet undergo maturation at mildly acidic pH.

Because no high enough resolution experimental structures of immature flavivirus particles at acidic pH are available (the reported structure of immature dengue virus at low pH is at 25 Å resolution[27]), we used the Alphafold3 server[44] to predict the potential molecular organization of a (prM/E)2 dimer, the building block or the herring-bone pattern. The resulting predictions essentially appose two prM/E protomers from the trimers of an immature particle to make a (prM/E)2 dimer that maintains the same head-to-tail contacts of the E dimer observed in mature virions. The FCS is not exposed in the prediction and becomes buried at the E dimer interface, albeit this region is predicted with low confidence (Fig. 4d). While these models may not capture the actual conformation, particularly since furin cleavage occurs at acidic pH and therefore the herringbone arrangement in the TGN is expected to expose the FCS – they point to regions near the FCS that may interact with the E protein at its dimer interface. For description purposes, we define the linker as the region connecting the globular pr head to the first ordered helix in the cryo-EM structure (labeled H1 in the Fig. 4d and supplementary Fig. 8), which must

somehow pass across the E dimer interface to connect with the prM helices anchored in the viral membrane.

To make the scissile bond accessible, the linker must unzipper at acidic pH. The reversibility between spiky and herringbone forms in MBFVs indicates that re-zippering occurs upon E dimer dissociation to form (prM/E)3 trimers, thereby retracting the scissile bond. In the case of TBFVs, if the herringbone pattern remains at neutral pH and the E dimer does not dissociate, re-zippering is likely to be obstructed, explaining the observed accessibility of the scissile bond on particles brought back to neutral pH.

To explore potential ways of tampering with FCS exposure by using reverse genetics, we considered the residues in the linker upstream the FCS, i.e, prM residues 79-84. Amino acid sequence alignment showed that in TBFVs, prM position 80 is always basic, Arg or Lys, whereas it is very variable in MBFVs, where it can also be glycine (Supplementary Fig. 5). In contrast, position 82 never has a basic residue, and it often has a negatively charged amino acid. The experimental structure of the prM/E dimer in the context of the spiky particle[36] shows that TBEV E82 makes a salt bridge with R87 of the FCS (Fig. 4d), contributing to partially neutralize the charged FCS. The AlphaFold prediction of the (prM/E)2 dimer places this segment of the prM linker running very close to a region of E that involves the fg loop, which has an insertion in TBFVs (residues 205-TVEHLP-210 in TBEV) compared to MBFVs (Supplementary Fig. 4a) on one protomer interacting with the motif 255-YN-256 at the end of β-strand j of the opposite E protomer. We therefore decided to combine two mutations in prM (K80G and E80R) with one mutation in E in this region. Analysis of the X-ray structures of the sE dimer at neutral pH[29], of the (pr/sE)2 dimer at acid pH[22], as well as the cryo-EM structures of the mature virion at neutral pH[41,42], showed an inter-protomer hydrogen bond between the main chain amine of Asn256 and the main-chain carbonyl of His208 in the fg loop. These structures also show that the side chain of Tyr255 packs against the ring of Pro210 across the dimer interface. Moreover, the amino acid sequence align-ment of E from multiple flaviviruses showed that the presence of the fg loop insertion correlates with the presence of asparagine at position 256 preceded by phenylalanine or tyrosine (Supplementary Fig. 4a). In the structures, Asn256 makes two hydrogen bonds with the main chain, one via its side chain amine and the other with its side chain carbonyl group. These two bonds position the Asn256 main chain with the right geo-metry to make an interprotomer hydrogen bond and simultaneously allow to the preceding Tyr255 to make an aromatic-proline CH/π interaction[45] with Pro210 across the dimer interface (Supplementary Fig. 4b). The AlphaFold predictions of the (prM/E)2 dimer place the region just preceding the FCS running in between Y255 and P210 (Supplementary Fig. 4c), such that they cannot contact each other, and the main chain interprotomer hydrogen bond cannot not form either. Because Asn256 appeared key to these interactions, we used the AlphaFold3 server to predict the structure of an N256A mutant TBEV sE dimer, which did not show the main chain inter-protomer hydrogen bond, and showed the ring of Tyr255 now packing at 90 degrees to the

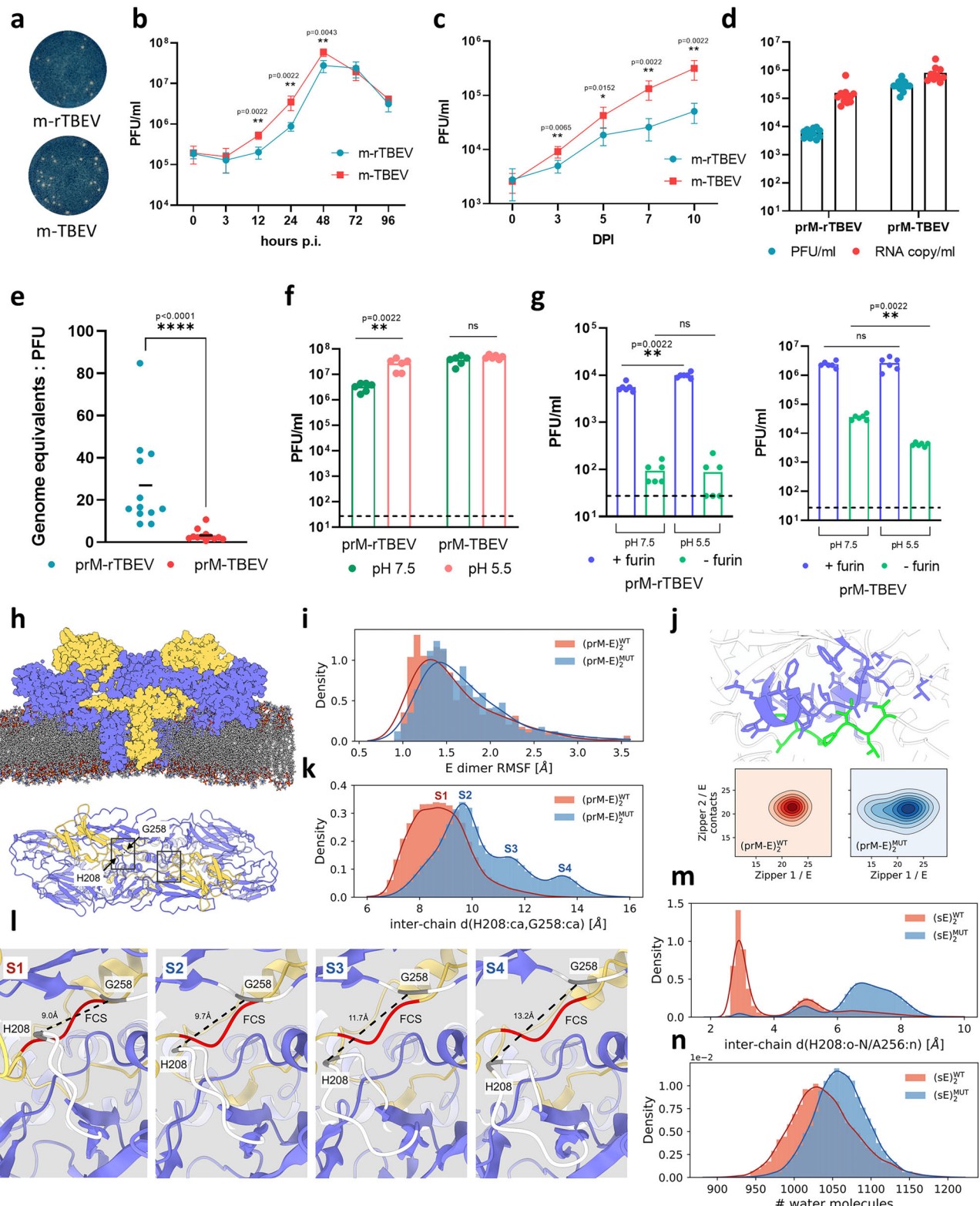

ring of Pro210 across the dimer interface (Supplementary Fig. 4b), in contrast to the prediction of a wt sE dimer, which, as expected, recapitulated the inter-protomer interactions observed experimentally.

### Reverse genetics to explore the phenotype of a mutant combining prM R80G/E82R and E N256A

We engineered a recombinant TBEV (m-rTBEV) incorporating the three mutations discussed above, in the hope that they may modulate FCS

accessibility: K80G and E82R in prM and N256A in E (Supplementary Fig. 10). Comparing the plaque morphology of m-rTBEV with m-TBEV revealed no differences in plaque size (Fig. 5a). Next, the replication kinetics in PS cells showed a peak at 48 hpi, and reached a titre of $2.7 \times 10^7$ for m-rTBEV, and $5.9 \times 10^7$ PFU ml$^{-1}$ for m-TBEV ($P < 0.01$) (Fig. 5b). Interestingly, the growth kinetics in the tick cell line IRE/CTVM19 showed a more pronounced attenuation at 10 dpi: titre of $5.1 \times 10^4$ PFU ml$^{-1}$ for m-rTBEV, compared to $3.2 \times 10^5$ PFU ml$^{-1}$ for

**Fig. 5 | Reverse genetics to explore the phenotype of a mutant TBEV and molecular dynamic simulations. a** Plaque morphology of mutant (m-rTBEV) and parental (m-TBEV) viruses. **b** Replication kinetics of both variants in mammalian PS cells and tick cells IRE/CTVM19 (**c**). **d** Plaque and RNA content in LoVo-derived immature prM-rTBEV and prM-TBEV after 48 hpi. **e**, Genomic RNA:PFU ratio for prM-rTBEV versus prM-TBEV. **f** pH sensitivity of virus samples pretreated with r-furin at acidic or neutral pH, followed by infection of PS cells. **g** Plaque assay of furin-treated virus samples to assess cleavage efficiency at different pH values. Data in **b**–**g** represent at least two independent biological replicates and are shown as mean ± SD. Statistics is done using the Mann–Whitney test; ns, $P > 0.05$, $*P < 0.05$, $**P < 0.01$, $***P < 0.001$, $****P < 0.0001$. **h** Schematic of MD simulation setup for $(prM-E)_2$ dimer in a lipid bilayer. Water molecules and ions are not shown (top panel). **i** Distributions of per-residue Root Mean Square Fluctuations (RMSF) across all residues of the E dimer. Raw normalized histograms are represented by bars, kernel density estimations as solid lines. **j** Two dimensional distributions of number of contacts formed between each of the two zippers in the dimer (green) and the neighboring protein E chain (violet), for $(prM-E)_2$WT (bottom panel, red) and $(prM-E)_2$MUT (bottom panel, blue) **k** Distributions of distances between the Cα atoms of residues H208 and G258 in $(prM-E)_2$, represented as in panel i. **l** Snapshots of representative configurations of $(prM-E)_2$WT (S1) and $(prM-E)_2$MUT (S2-S4). The protein E loop D203-T211 as well as residues L257 and D259 are coloured in white, residues H208 and G258 in gray, and the FCS on prM in red. Black dashed lines indicate the distance between the Cα atoms of residues H208 and G258. **m** Distributions of distances between the carbonyl oxygen of residue 208 and the amide nitrogen of residue 256 in $(sE)_2$, represented with red and blue colors indicating $(sE)_2$WT and $(sE)_2$MUT, respectively. **n** Distribution of number of water molecules in the first coordination shell of atoms at the interface of the $(sE)_2$ dimer, represented as in panel m. Colors indicate WT (red) and mutant (blue) variants throughout panels (**i**–**n**).

m-TBEV ($P < 0.01$) (Fig. 5c). Overall, m-rTBEV exhibited slightly reduced growth ability compared to m-TBEV in mammalian cell lines, but a greater attenuation when infecting tick cells.

Next, we infected LoVo cells to produce an immature variant of rTBEV (prM-rTBEV). At 48 hpi, prM-rTBEV reached a titre of $6 \times 10^3$ PFU ml$^{-1}$, compared to $3 \times 10^5$ PFU ml$^{-1}$ for prM-TBEV (Fig. 5d). The RNA levels (Fig. 5d) resulted in a genome equivalent:PFU ratio of ~27 for prM-rTBEV compared to ~2 for prM-TBEV ($P < 0.0001$) (Fig. 5e). These data indicated a shift in the specific infectivity of the recombinant immature variant towards that of MBFV. To assess the pH dependence of prM-rTBEV, the virus was pretreated with r-furin and then PS cells were infected. The virus titre of prM-TBEV barely changed, while the virus titre of prM-rTBEV was lower at neutral pH compared to acidic pH, suggesting that furin cleavage was more efficient with prM-rTBEV at acidic pH ($P < 0.01$) (Fig. 5f). To obtain more detailed information, we incubated the virus with r-furin and then immediately performed a plaque assay. Immature prM-rTBEV showed a slightly reduced efficiency of furin cleavage at pH 7.5 compared to an acidic pH ($P < 0.01$), indicating a change in sensitivity similar to that of MBFV. Interestingly, no differences were observed between prM-TBEV samples incubated at acidic and neutral pH, but cleavage by endogenous furin during the plaque assay was more efficient at neutral pH compared to acidic pH ($P < 0.01$) (Fig. 5g). This trend in cleavage efficiency was opposite to our observations with prM-rTBEV. Overall, these results suggest that prM-rTBEV has generally similar maturation properties to prM-USUV (MBFV). Taken together, our data indicate that the introduced mutations alter the pH-dependent behavior of immature virions. We show that immature TBEV virions are readily cleaved by furin at the surface of target cells, whereas those of MBFVs are not. Our data indicate therefore that in the mutant, the E dimers may dissociate upon re-exposure to neutral pH in the absence of furin, making the change with pH as in MBFVs, thereby allowing prM-zippering to retract the scissile bond.

**Molecular dynamics simulation support destabilization of the sE dimer in the mutant**

To assess the effect of mutations on the stability of the $(prM/E)_2$ dimer, we first built a comparative model of the triple mutant $(prM/E)_2$MUT dimer starting from the AF3 model of the wild type $(prM/E)_2$WT (Methods). We then performed 1 μs-long all-atom, explicit solvent molecular dynamics simulations (MD) of $(prM/E)_2$WT and $(prM/E)_2$MUT immersed in a heterogeneous lipid bilayer with composition mimicking the mammalian endoplasmic reticulum membrane (Fig. 5h, top panel). The distributions of Root Mean Square Fluctuations (RMSF) indicate that, overall, the three mutations have a weak destabilizing effect on the E dimer (Fig. 5i). However, both systems remain largely stable in the microsecond timescale, with median RMSF calculated on the entire $(prM/E)_2$ dimer equal to 1.5 Å and 1.7 Å for $(prM/E)_2$WT and $(prM/E)_2$MUT, respectively. Furthermore, the weak destabilization of the E dimer does not appear to affect the prM zipper, which remains

inserted in the E αA-αB groove in both wild-type and mutant simulations (Fig. 5j).

We then focused on the protein E fg loop, between residues D203 and T211, which is situated in close proximity of the FCS and in the initial $(prM/E)_2$WT AF3 model appears to block FCS exposure to solvent and ultimately cleavage by furin (Fig. 5h, bottom panel). The MD simulation of $(prM/E)_2$WT indicates that this loop is firmly locked in the position observed in the AF3 model (Fig. 5k, in red and Fig. 5l, state S1). On the other hand, in $(prM/E)_2$MUT the loop becomes more dynamic, sampling conformations that could allow the FCS to extend in solution (Fig. 5k, in blue and Fig. 5l, states S2-S4). For comparison, we also performed MD simulations of the wild-type, soluble E dimer $(sE)_2$WT (PDB code 1SVB) and of the N256A single mutant $(sE)_2$MUT. This mutation disrupted the backbone inter-protomer hydrogen bond between residues 208 and 256 (Fig. 5m) and led to the same increased dynamics of the fg loop observed in the triple-mutant $(prM/E)_2$MUT (Supplementary Fig. 9). Furthermore, residues at the interface of the $(sE)_2$ dimer appeared less buried, and more solvent exposed upon mutation (Fig. 5n). Altogether, our MD simulations support the hypothesis that the three mutations of $(prM/E)_2$, - and most likely the N256A mutation by itself - destabilize the E dimer interface, facilitating FCS retraction by zippering upon E dimer dissociation at neutral pH, as in MBFVs.

## Discussion

MBFVs and TBFVs have long been thought to share fundamental molecular and structural properties, with correspondingly similar replication and maturation strategies within host cells[37]. Yet we here reveal crucial disparities between the maturation processes of these two virus groups. Previous studies of immature flaviviruses have predominantly focused on structural characteristics, focusing mostly on the properties of the neutral pH spiky immature particle[27,28,46,47]. In contrast, here we explored functional aspects of an intermediate TBFV variant - an immature virion which, unlike the spiky particle, can be processed by furin at neutral pH.

Our study revealed substantial differences in infectivity between immature TBFVs and MBFVs. The infectivity of mature versus immature TBFVs differed by a maximum of 10-fold, while this discrepancy was significantly greater (1,000- to 10,000-fold) for MBFVs. Immature DENV, a typical MBFV, was previously described as infective in the presence of disease-enhancing antibodies (ADE), which enable otherwise non-infectious immature virus to enter cells[48–50]. Here, we found that immature TBFVs were infectious without ADE.

We confirmed that furin cleavage was necessary in all cases, consistent with previous findings for TBEV and other flaviviruses[22]. Moreover, we found that furin cleavage was not limited to the virus-producing cell, but also occurred on target cells during virus entry, supporting earlier hypotheses that final virus maturation might occur upon host cell entry under specific conditions[40,51]. For example, the

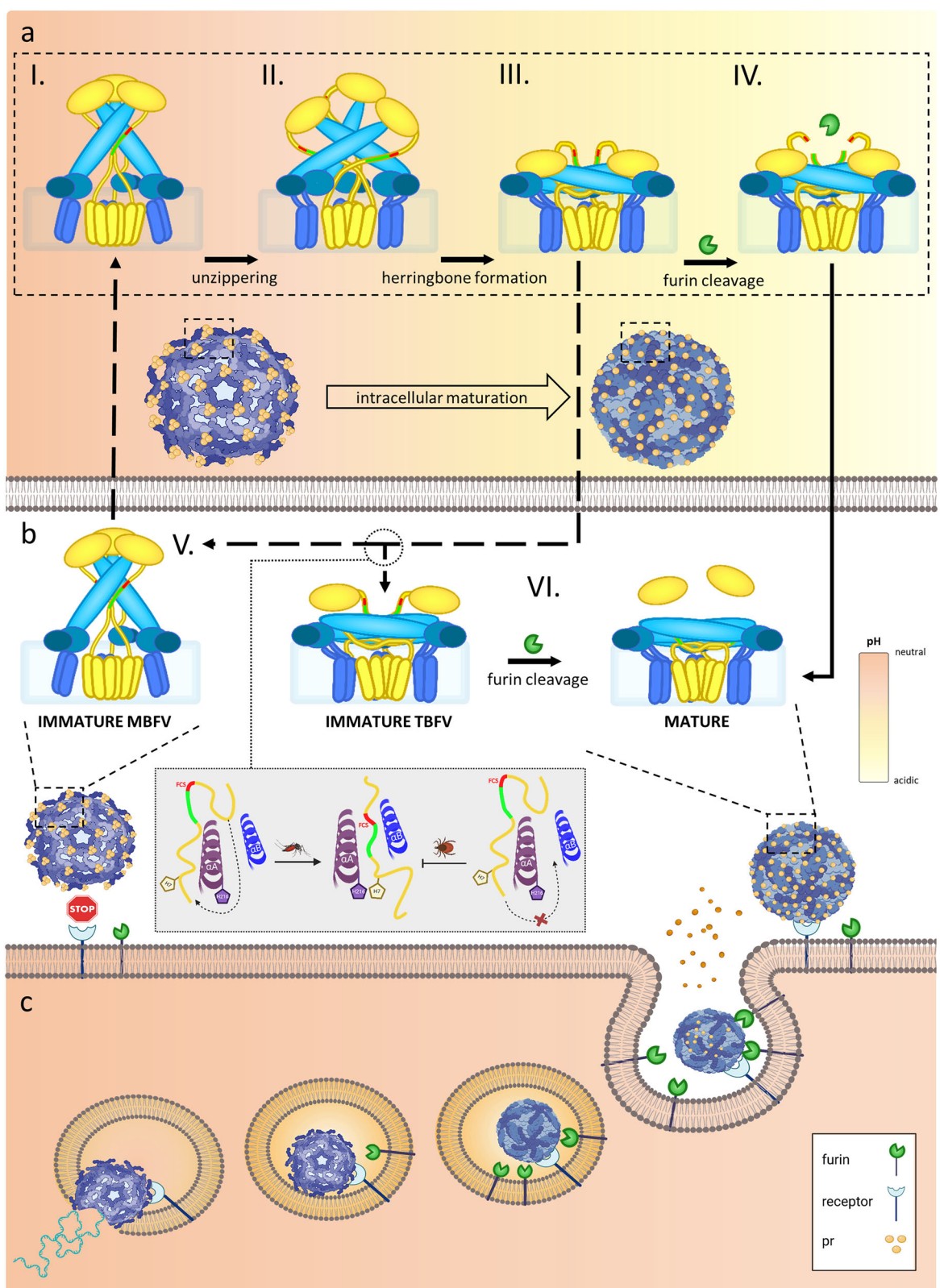

insect-specific flavivirus Binjari virus circulates in an immature form and undergoes furin cleavage only upon subsequent host cell entry[52].

Furin cleavage typically takes place in the acidic environment of the Golgi, where the pr portion of prM remains bound within a basic cavity at the E dimer interface. Exposure to neutral pH extracellular milieu induces a change in the glycan loop, which closes the cavity with concomitant pr ejection. This structural rearrangement primes the virion for activation of its membrane fusion machinery, which is triggered upon re-exposure to acidic pH in the endosome of a newly infected cell.[14,24,25]. Thus, prM functions as a chaperone, preventing virions from engaging in fusion within the host cell's acidic environment before release[23].

The observed reversibility of the transition from spiky to herringbone forms in MBFVs in the absence of furin cleavage[27] implies E

**Fig. 6 | Schematic representation of maturation differences during the infectious cycle of TBFVs and MBFVs.** Schematic representation of flavivirus maturation process upon assembly of new particles in an infected cell, reflecting current understanding based on structural studies, experimental observations, and model-based predictions. **a–I** Immature particles, which bud at neutral pH into the ER lumen, contain 60 spikes formed of (prM/E)₃ trimers depicted with prM in yellow and E in blue. PrM has a globular pr head masking the fusion loop in E, and a long linker that connects to its transmembrane domains. The linker contains a zippering element (green), which inserts at neutral pH into a non-polar groove in E, thereby protecting the adjacent FCS (in red) from exposure and cleavage. **a–II** During transport through the GA, the particle is exposed to increasingly acidic pH, which triggers (E/prM)₃ dissociation and ejection of the zipper from the E groove, effectively exposing the FCS. **a–III** The E/prM protomers reorganize to make 90 (prM/E)₂ dimers in a smooth herringbone lattice, with the pr moiety of prM still masking the fusion loop but with the linker unzipped and the FCS exposed. **a–IV** The presence

of furin (green pacman) results in cleavage and maturation of the particle. MBFVs and TBFVs released from furin-deficient cells (into the extracellular environment with neutral pH – panel B) show different behavior: while the former reverts to the spike form adopted in the ER, resulting in re-zippering and re-formation of (prM/E)₃ trimers (**b–V**) this is not the case for TBFVs. **b–VI** In this case, the more stable interface between E protomers of the (prM/E)₂ dimer prevents the zipper from re-entering the E groove, which is located beneath the dimer, facing the viral membrane. The FCS thus remains exposed at the surface of smooth particles and furin cleaves it at neutral pH, in striking contrast to MBFVs. In TBFVs, this cleavage can occur at the host cell membrane upon infection, leading to the release of the cleaved pr fragment and subsequent endocytosis (**b–VI** and **c**). During endocytosis, the decreasing pH activates the fusion loop, facilitating fusion with the endosomal membrane and initiating a new infectious cycle (**c**). Selected graphics were created in BioRender. Ruzek, D. (https://BioRender.com/4n4b27g).

dimer dissociation, in turn allowing prM re-zippering, as observed in the available structures of spiky particles[24,36,52,53]. The irreversibility of the same transition for TBFVs[14] suggests that the E dimers remain formed at neutral pH and obstruct re-zippering. The emerging picture is thus that once formed by exposure to the acidic environment of the Golgi, the herringbone pattern of E dimers remains when the immature TBFV particle is exposed back to neutral pH, and therefore that immature virions circulate as smooth particles, in contrast to the spiky immature MBFVs, although this remains to be addressed and confirmed by direct structural data, for example from cryo-EM. The absence of the cryo-EM data on these structures represents one of the limitations of the present study. Upon furin cleavage, the newly form N-terminal segment of M can snap back in the context of E dimers, as shown by the available structures of mature virions[41,42]. Our finding that immature TBFVs are fully infectious shows that the smooth immature virions are not hindered in their ability to bind to host cells. We propose that, at neutral pH, the globular pr domain of prM is loosely tethered to the virion surface (Fig. 6) rather than firmly anchored, as it is under acidic conditions[22]. We hypothesize that the tether extends out by stretching of the M loop (labeled in Suppl. Figure 5) into an extended conformation connecting to the αH1 helix on the membrane, projecting the zipper region outside the E dimer so that the scissile bond is accessible for cleavage.

Although the glycan loop in domain I of E was previously proposed to stabilize the E dimer by relaying the pr moiety and promoting irreversible conformational changes[22], our findings indicate that this relay is not sufficient by itself. We identified a critical role for the fg loop (residues 203–211) and the aromatic-asparagine motif (φN, where φ represents phenylalanine or tyrosine) in β-strand j (Supplementary Fig. 4). These two elements, which are unique to TBFVs among all flaviviruses examined, make key stabilizing inter-protomer interactions across the E dimer interface: a direct inter-chain hydrogen bond and an aromatic-proline CH/π bond conserved in TBFVs and absent on MBFVs and other flaviviruses. Our analysis highlights that disruption of this module via the N256A mutation destabilizes the E dimer, likely allowing the reversion to the spiky form at neutral pH, similar to MBFVs, with concomitant FCS retraction.

The differential properties displayed by immature flaviviruses identified here may derive from different evolutionary pressures within their hosts and vectors, leading to divergent maturation strategies and biological behaviour[54]. A comprehensive analysis of TBFVs is imperative to elucidate their unique characteristics and evolutionary adaptations, offering valuable insights into their infectivity, transmission, and pathogenicity.

In conclusion, our results highlight critical differences between immature MBFVs and TBFVs. They challenge the dogma positing that immature flaviviruses exhibit uniformly low infectivity and enhance our understanding of flavivirus transmission dynamics and pathogenicity. These data have the potential of guiding targeted antiviral

strategies against flavivirus infections while elucidating the evolutionary pressures shaping these virus groups.

## Methods

### Virus strains

TBFV were represented by TBEV strain Hypr (of the European subtype), LIV (strain LI/31) and LGTV (strain TP-21), which were all provided by the Collection of Arboviruses, Institute of Parasitology, Biology Centre of the Czech Academy of Sciences, Ceske Budejovice. MBFV used in this study included ZIKV (Brazilian strain Paraiba_01), kindly provided by Prof. Paolo M. de A. Zanotto, University of São Paulo, Brazil; USUV (Eu1 lineage strain 200/TM/10), provided by the Collection of Arboviruses, Institute of Parasitology, Biology Centre of the Czech Academy of Sciences[55]; and WNV (strain 13-104, representative of genomic lineage 2, isolated in the Czech Republic), kindly provided by Prof. Zdenek Hubalek and Dr. Ivo Rudolf, Institute of Vertebrate Biology of the Czech Academy of Sciences[56].

### Mammalian and tick cell lines

Porcine kidney stable PS cells (National Reference Laboratory for Cell Cultures, National Institute of Public Health, Prague) were grown in Leibovitz' L-15 medium (L-15). Golden hamster BHK-21 (ATTC CCL-10) and African green monkey Vero cells (ATCC CCL-81) were grown in Dulbecco's modified Eagle's medium (DMEM). Furin-deficient human LoVo cells (human colon carcinoma, ATCC CCL-229) were grown in Ham's F12 medium. Cells were cultured in media supplemented with foetal bovine serum (3% in L15, 10% in DMEM, or 20% in Ham's F12), 100 U ml⁻¹ penicillin, 100 µg ml⁻¹ streptomycin, and 2 mM glutamine (Sigma-Aldrich, Prague, Czech Republic), at 37 °C in 5% $CO_2$ (except for PS cells, which were cultured without extra $CO_2$). The tick cell line IRE/CTVM19[57] (derived from *Ixodes ricinus*, Tick Cell Biobank, University of Liverpool, UK) was maintained at 28 °C without extra $CO_2$, in L15 medium supplemented with 10% foetal bovine serum and 5% tryptose phosphate broth, as previously described[58].

### Production of prM-containing immature particles

Immature prM-containing virus particles were produced using LoVo cells, which are deficient in furin protease[59], and thus provide an easy-to-use platform for immature virus production without affecting the pH within the maturation pathway. For propagation, cells were seeded in 96-well plates (approximately $3 \times 10^4$ cells per well), and incubated at 37 °C under 5% $CO_2$ for 24 h, to form a confluent monolayer. Then these cells were infected with viruses at a multiplicity of infection (MOI) of 2. After 12 h, the virus suspension was removed, the cells were washed three times with phosphate-buffered saline (PBS), and fresh medium was added. At 48 h post-infection (hpi), the virus was harvested, and viral titre was determined by plaque assays, or viral RNA

copies were quantified by quantitative PCR with reverse transcription (RT-qPCR), as described below.

## Plaque assay

Plaque assays were performed as previously described[60], with slight modifications. Briefly, viruses were diluted 10-fold in 24-well tissue culture plates, and $1-1.5 \times 10^5$ PS or Vero cells were added to each well. After 4 h of incubation, the wells were covered with 1.5% (w/v) carboxymethylcellulose (CMC) in L-15 (PS cells for TBEV, LGTV, LIV, and USUV) or DMEM (Vero cells for ZIKV and WNV) medium. After five days of incubation at 37 °C, infected plates were washed with PBS, and cell monolayers were stained with naphthol blue-black. Viral titre was expressed as plaque-forming units (PFU) per ml.

## RNA isolation, RT-qPCR, RT-PCR, and sequencing

Viral RNA was isolated from culture supernatants using the QIAmpViral RNA mini kit (Qiagen, Germantown, MD, USA) following the manufacturer's instructions. RT-qPCR was performed on a LightCycler 480 II in a 96-well plate block (Roche, Basel Switzerland), using the Advanced Kits for Tick-borne Encephalitis, Zika Virus, or West Nile Virus (Genesig, Germantown, MD, USA) or the Usutu Virus Kit (#FR439; Genekam Biotechnology AG, Duisburg, Germany). Lyophilized OneStep qRT-PCR master mix (Oasig) was used, following the manufacturer's instructions. For TBEV, ZIKV, and WNV, the copy numbers per µl were calculated from calibration curves based on standards provided by the manufacturer. For USUV, the calibration curve was developed using serial dilutions of RNA isolated from a virus stock with a known titre in PFU ml$^{-1}$; therefore, the qRT-PCR results for USUV were presented in relative PFU per ml (rPFU ml$^{-1}$). For sequencing, the isolated RNA was reverse-transcribed using a QIAGEN OneStep Ahead RT-PCR Kit (Qiagen, Germantown, MD, USA). Amplified products were separated in 1% agarose gel, and purified using the Wizard SV Gel and PCR Clean-Up System (Promega, USA). Sanger sequencing was performed by the commercial provider Eurofins Genomics.

## Plaque-reduction assay

PS cells were seeded in 6-well tissue culture plates ($0.8-1 \times 10^6$ cells per well), and incubated for 24 h. Confluent cells were pretreated for 2 h with the furin inhibitor Decanoyl-RVKR-CMK (Sigma Aldrich), and then infection was performed using 50 PFU per well, at 37 °C for 1 h. Next, the unbound virus was removed, the cells were washed three times with PBS, and fresh medium with a CMC overlay was added. After incubation at 37 °C for five days, infected plates were washed with PBS, and cell monolayers were stained with naphthol blue-black as described for the plaque assay.

## Virus in vitro furin cleavage assay

The prM-TBEV sample was concentrated using Amicon ultra-centrifugal 100-kDa filters, and medium-buffer exchange was performed by centrifugation (4500 rpm, 4 °C, 7 min; TX-400 rotor, 75003629; Thermofisher Scientific). Next, the medium was changed to Tris-maleate buffer[14] (TMB, 50 mM) containing CaCl$_2$ (10 mM), with the pH adjusted to 5.5 or 7.5. For furin in vitro cleavage, 1 µl of recombinant human furin (r-furin, #SRP6274; Sigma Aldrich) was added to the 100-µl reaction, followed by incubation at 37 °C for 2 h. Next, LoVo cells in TMB were incubated with the virus (MOI 0.1) for 4 h. Then the unbound virus was removed, the cells were washed three times with PBS, and fresh medium was added. Finally, at 48 hpi, the viral progeny was harvested for further analysis. Infection with prM-USUV was performed using PS cells, with the same in vitro cleavage procedure as described above.

## Fluorescent infectious assay

Recombinant mCherry-expressing TBEV was used to visualize viral infection[61]. PS cells were cultured in 96-well plates, and infected as described above. At 48 hpi, nuclei were stained with Hoechst 33342 (Invitrogen), following the manufacturer's recommended protocol. Immunofluorescence of prM-TBEV and prM-USUV infection was visualised after fixation in ice-cold methanol-acetone mixture (1:1) with primary anti-flavivirus protein E antibodies (1:250; Sigma Aldrich) and then with secondary antibodies AlexaFluor 488 (1:1000; Thermofisher Scientific) and DAPI (1:2000; Sigma Aldrich) to stain cell nuclei. Images were acquired using either an Olympus IX81 epifluorescence microscope (equipped with Olympus 10× UPLFLN lenses and a Hamamatsu OrcaR2 camera, controlled by Olympus Xcellence software) or the ImageXpress Pico automated imaging system with CellReporterXpress software (Molecular Devices, USA) for image acquisition and analysis. Raw images were processed using ImageJ/Fiji software.

## Western blot analysis

To investigate prM cleavage by furin, we performed western blot analysis. Virus samples used for western blot analysis were produced in medium mixture DMEM:HAM F12 (1:1) supplemented with a reduced FBS concentration (2%), and concentrated using Amicon ultra-centrifugal 100-kDa filters. Laemmli sample buffer was mixed 1:3 with the virus sample, and incubated for 5 min at 95 °C. After cooling on ice, samples were ready for downstream analysis. Proteins were separated by standard SDS-PAGE, and then transferred to an Immobilon®-P PVDF Membrane (Millipore). Blots were blocked overnight at 4 °C in blocking buffer comprising PBS, 0.05% Tween 20, and 2% Amersham ECL Prime Blocking Reagent (Cytiva). Next, the blots were stained with primary antibodies against TBEV E protein (mouse monoclonal antibody 1493[62] diluted 1:2000) or M protein (in-house rabbit polyclonal serum[42] diluted 1:500), overnight at 4 °C, followed by the secondary antibodies anti-mouse IgG-HRP (Invitrogen AB_228307) or anti-rabbit IgG-HRP (Invitrogen AB_228341) for 1 h at room temperature. Finally, blots were visualized on an Amersham™ Imager 680 (GE Healthcare).

## Mouse infection

We evaluated the infectivity of prM-containing viruses in 6-week-old female BALB/c mice (ENVIGO RMS). Female mice were used, as they are the most frequently used model in TBEV research[63–68]. All animal procedures were conducted in a biosafety level 3 (BSL-3) facility. Mice were housed in individually ventilated cages under controlled environmental conditions with a 12-h light/dark cycle. Animals had ad libitum access to food and water and were monitored daily for health status and signs of disease or distress.

Six groups of mice were inoculated s.c. with TBEV or prM-TBEV strain Hypr as follows: group 1 ($n = 5$, single experiment), TBEV $10^3$ genomic equivalents (gen. eq.) per mouse; group 2 ($n = 5$ or 6 mice per group, total 11 mice per group in two independent experiments), TBEV $10^2$ gen. eq. per mouse; and group 3 ($n = 5$, single experiment), TBEV $10^1$ gen. eq. per mouse, respectively. Groups 4–6 were infected with the equivalent doses of the prM-TBEV variant. For comparison with MBFV, WNV strain 13-104 was used and the experimental set-up and mouse groups were the same as in the TBEV experiment. Over a 28-day experimental period (all mice), the survival rates and symptoms were monitored daily. Symptoms were evaluated using the following clinical scores 1 = healthy, 2 = piloerection, 3 = hunched back, 4 = paralysis, and 5 = death. Changes in body weight upon infection were monitored for 14 days ($n = 6$ mice per group, 24 mice in total, single experiment). All mice exhibiting a clinical score of 4 were humanely terminated (by cervical dislocation) immediately upon detection of symptoms.

Blood was taken 3 dpi from the tail vein to evaluate the virus titre in the serum (n = 6 mice per group, single experiment). After dissection of the brain (n = 5 per group, single experiment) on day 8, tissue samples were homogenized in DMEM medium (+ 10% FBS) to obtain a

20% suspension. After centrifugation (5000 rpm, 4 °C, 10 min; TX-400 rotor, 75003629; Thermofisher Scientific), the supernatant from suspension was used for the plaque assay.

## Protease fluorescent assay

Four internally quenched fluorescent substrates—Dabcyl-EGSRSRRSVL-Edans, Dabcyl-EGSRTRRSVL-Edans, Dabcyl-HSRRSRRSLT-Edans, and Dabcyl-HSKRSRRSIA-Edans—were synthesized without C-terminal Edans, using a standard solid phase peptide synthesis protocol[69], with 2-chlorotrityl chloride resin support. The N-Dabcylated side chain protected peptides were released from the resin using a mixture of acetic acid/2,2,2-trifluoroethanol/dichloromethane (1:1:3) for 2 h, and then were evaporated and dried. Next, the Edans was introduced via Edans acid sodium salt (1.5 eq. to peptide) and PyBOP (2.0 eq. to peptide) in the presence of DIPEA (3 eq. to peptide) in DMF as a solvent, overnight. After solvent evaporation, side chains were deprotected using a mixture of TFA/triisopropylsilane/water (95:2.5:2.5) for 1 h, and then liquids were evaporated. The residues were purified by preparative RP HPLC, and characterised by LC/MS-ESI: Dabcyl-EGSRSRRSVL-Edans [M + H]$^+$ 1645.8, Dabcyl-EGSRTRRSVL-Edans [M + H]$^+$ 1659.8, Dabcyl-HSRRSRRSLT-Edans [M + H]$^+$ 1754.8, and Dabcyl-HSKRSRRSIA-Edans [M + H]$^+$ 1696.9.

Lyophilized human recombinant furin protease produced in Hi-5 insect cells (PeproTech) was diluted to 1 mg ml$^{-1}$ using water, according to the manufacturer's instructions. The assay buffer of an appropriate pH contained 25 mM acetic acid, 25 mM 2-[morpholino] ethanesulfonic acid (MES), 25 mM glycine, and 1 mM CaCl$_2$[70]. Fluorescence measurements were performed in 96-well Greiner chimney black plates. All assays were started by adding furin to a final concentration of 10 ng mL$^{-1}$ in the 100-µl reaction volume. The time-dependent increase of fluorescence upon enzyme addition was monitored for 30 min, with a kinetic cycle of 1 min, using a Tecan Spark plate reader with an excitation wavelength of 336 nm and emission wavelength of 490 nm. All measurements were made in triplicate.

## Quaternary structure analysis

For both TBEV and USUV, models of the prM/E complexes (containing the complete protein sequences of the prM and E proteins) were made using AlphaFold2[71] and AlphaFold3[44]. These were used to generate the quaternary structure model of a dimer of heterodimers, based on the PDB code templates 7Z51, 5O6A, 7LCH, and 7QRE[22,42,72,73]. Structural analysis of the models was performed in PyMOL 2.5.4 and USCF ChimeraX.

## Site-directed mutagenesis and transfection

The reverse genetics system used was based on the generation of three infectious subgenomic overlapping DNA fragments encompassing the whole TBEV genome (strain Hypr), as previously described[74]. Fragments I and III were flanked by the human cytomegalovirus promoter (pCMV) and hepatitis delta ribozyme, followed by the simian virus 40 polyadenylation signal sequence (HDR/SV40pA). All three fragments were synthesized and individually cloned into pUC57 or pC11 vectors (GenScript, Piscataway, NJ, USA).

To produce genomic fragments with mutations in prM and E proteins, fragment I was used as a template for amplification with modified primers (Supplementary Fig. 10, and Supplementary Table 1). This enabled production of two sub-fragments, Ia and Ib, each with overlapping homology arms. PCR and fragment purification were performed as previously described[61].

For transfection, BHK-21 cells were seeded in a 24-well plate at a density of $3 \times 10^5$ cells (400 µl) per well, and incubated overnight. Cell transfection was performed using an equimolar mixture of the four DNA fragments. A DNA-lipid complex was prepared using X-tremeGENE Transfection Reagent and 200 µl Opti-MEM (Thermofisher Scientific), followed by a 15-min incubation at room temperature. The entire mixture was added drop-wise to the well with the cell monolayer, and then incubated for 4 days. Then the supernatant was collected, and viral titre was determined by plaque assay. Finally, mutagenesis was confirmed by sequencing.

## Statistics, reproducibility, and graphics

GraphPad software Prism v.8 was used for statistical analysis. In all figures, data were plotted as means with error bars denoting the SD. All statistical tests were performed using the Mann-Whitney nonparametric test. In animal experiments, the survival rates were statistically evaluated using the log-rank Mantel-Cox test. The significance threshold was set at $P < 0.05$. Figures 1a; 2a, h; 4a; 6a–c and Supplementary Fig. 10 were created or partially created with BioRender.com under the publication license.

## Molecular Dynamics simulations

**Initial models of the mutant (prM/E)$_2$ and (sE)$_2$ dimers.** Comparative models of the triple mutant (prM/E)$_2$$^{MUT}$ dimer (prM: K80G, E82R; E: N256A) were constructed using the AF3 model of the wild type (prM/E)$_2$$^{WT}$ dimer as template. Distance restraints were added to ensure the formation of the 18 disulfide bonds present in the template. For each construct, 1000 comparative models were built using MODELLER v. 10.1[75] and ranked based on the normalized DOPE score[76]. The top scoring (prM/E)$_2$$^{MUT}$ model was used as starting point of the MD simulations, as described in the following section. The same procedure was applied to build comparative models of the N256A single mutant (sE)$_2$$^{MUT}$ starting from the X-ray structure of the soluble E dimer (PDB code 1SVB).

**Details of the MD simulations: setup, equilibration, and production.** The (prM/E)$_2$$^{WT}$ AF3 model and (prM/E)$_2$$^{MUT}$ comparative model were first oriented in an implicit lipid bilayer parallel to the $xy$ plane using the PPM 3.0 webserver[77] and then used as input to the CHARMM-GUI server[78]. The two systems were embedded in a symmetric lipid bilayer (Fig. 5h) with composition mimicking the mammalian endoplasmic reticulum membrane: 60% POPC, 25% POPE, 5% POPS, 5% POPI, and 5% cholesterol (CHOL). The systems were then solvated in a triclinic box of initial x-y-z dimensions equal to ~16.2 nm * 16.2 nm * 13.2 nm. Potassium and chloride ions were added to ensure charge neutrality at a salt concentration of 0.15 M. The total number of atoms was ~320,000. Additional details of the systems are reported in Supplementary Table 2. The CHARMM36m force field[79] was used for protein, lipids, and ions and the mTIP3P model[80] was used for the water molecules. The standard CHARMM-GUI protocol, which consisted of energy minimization followed by multiple consecutive short simulations in the NVT and NPT ensembles, was used to equilibrate the systems. During these equilibration steps, harmonic restraints on the positions of the lipid and protein atoms were gradually switched off. The final step was extended to 50 ns to allow additional time for lipid equilibration. For each system studied, production simulations were performed in the NPT ensemble for 1 µs. The temperature T was set at 303 K and the pressure P at 1 atm using the Bussi-Donadio-Parrinello thermostat[81] and a semiisotropic Parrinello-Rahman barostat[82], respectively. A time step of 2 fs was used together with LINCS constraints on h-bonds[83]. van der Waals interactions were gradually switched off at 1.0 nm and cut off at 1.2 nm; the particle-mesh Ewald method was used to calculate electrostatic interactions with cutoff at 1.2 nm[84]. A similar protocol was used to perform the MD simulations of the soluble (sE)$_2$$^{WT}$ and (sE)$_2$$^{MUT}$ dimers. All simulations were performed with GROMACS v. 2022.5[85].

**Details of the analysis.** To evaluate the stability of the E dimer in (prM/E)$_2$$^{WT}$ and (prM/E)$_2$$^{MUT}$, we first aligned each trajectory to the initial conformation and then calculated for each residue of the E dimer the Root Mean Square Fluctuations (RMSF) using the *gmx*

*rmsf* tool provided by GROMACS. Finally, we calculated the distributions of RMSF along the two chains of the E dimer (Fig. 5i). To evaluate the overall stability of the $(prM/E)_2$ dimers, we also calculated the RMSF on the entire systems. To monitor the positions of the two zippers (prM residues 90-96), we calculated, for each frame of the trajectory and independently for each prM chain, the number of contacts between the zipper and residues 214-227/257-272 of the neighboring protein E chain (Fig. 5j, top panel). Two residues were defined in contact if the distance between their Cα atoms were lower than 8 Å. We then computed a two-dimensional distribution of contacts formed by one zipper versus the other to assess the symmetry between the two prM chains (Fig. 5j, bottom panels). To monitor the position of the protein E loop encompassing residues D203 to T211, we calculated for each frame of our trajectories the distance d(H208, G258) between the Cα atoms of H208 and G258, which faces H208 on the opposite E dimer chain (Fig. 5h, bottom panel). We then calculated the distribution of d(H208, G258) across the two chains of the E dimer (Fig. 5k). The same analysis was performed for the MD simulations of the soluble $(sE)_2^{WT}$ and $(sE)_2^{MUT}$ dimers (Supplementary Fig. 9). To monitor the formation of the backbone hydrogen bond between residues 208 and 256, we calculated the distance between the carbonyl oxygen of residue 208 and the amide nitrogen of residue 256 in both $(sE)_2^{WT}$ and $(sE)_2^{MUT}$ (Fig. 5m). The zipper, protein E loop, and hydrogen bond analysis were performed using MDAnalysis v. 2.3.0[86]. Finally, we quantified the buriedness of the residues at the interface of the $(sE)_2$ dimer by calculating the total number of water molecules within the first solvation shell of all the heavy atoms belonging to these residues (Fig. 5n). Interfacial residues were defined using a distance cutoff of 8 Å in the X-ray structure of $(sE)_2^{WT}$ (PDB code 1SVB). The number of water molecules was then computed using the COORDINATION action of the PLUMED open source library[87] with a distance cutoff equal to 3.5 Å.

## Ethics statement

The animal experiments complied with all relevant European Union guidelines for work with animals, and were conducted in accordance with Czech national law guidelines on the use of experimental animals and protection of animals against cruelty (Animal Welfare Act No. 246/1992 Coll.). The protocol was approved by the Committee on the Ethics of Animal Experimentation of the Institute of Parasitology, and the Departmental Expert Committee for the Approval of Projects of Experiments on Animals of the Czech Academy of Sciences (permit no. 111/2020).

## Inclusion & ethics

All collaborators of this study have fulfilled the criteria for authorship required by Nature Portfolio journals have been included as authors, as their participation was essential for the design and implementation of the study. Roles and responsibilities were agreed among collaborators ahead of the research.

## Reporting summary

Further information on research design is available in the Nature Portfolio Reporting Summary linked to this article.

## Data availability

All data supporting the findings of this study are available within the article and its supplementary files. Any additional requests for information can be directed to, and will be fulfilled by, the corresponding author. Accession codes for the structural data used in this study can be found here: 1SVB, 7Z51, 5O6A, 7LCH, 7QRE and 8PUV. Source data are provided with this paper.

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

## Acknowledgements

This study was supported by the Czech Science Foundation (grant No. 23-07160S, awarded to D.R.), the EEA and Norway Grants Fund for Regional Cooperation (project number 2018-1-0659, awarded to D.R.); and the National Institute of Virology and Bacteriology (Programme EXCELES, ID Project No. LX22N-PO5103), funded by the European Union—Next Generation EU. This study was also supported by Charles University Grant Agency (project no. 50121). L.B.S. was supported by the Wellcome Trust (grant no. 223743_Z_21_Z). A.K.Ö was supported by the Swedish Research Council (grant no. 2020-06224). We thank Jan Konvalinka and Milan Kožíšek for fruitful discussions.

## Author contributions

J.H., D.R. and F.A.R. conceived and designed the study and wrote the manuscript. F.A.R. and M.B. performed structural analyses, molecular dynamics simulations and wrote relevant parts of the manuscript. J.H., J.S., M.M., P.B., P.N., M.H., T.M., E.R., L.E., A.F., M.B., A.Ö, and A.C. performed the analyses. L.B.-S. contributed essential materials. All authors reviewed the manuscript.

## Competing interests

The authors declare no competing interests.
