## [Transparent Peer Review file · Nature Communications]

Irreversible furin cleavage site exposure renders immature tick-borne flaviviruses fully infectious

Corresponding Author: Professor Daniel Ruzek

Editorial Note: Parts of this peer review file have been redacted as indicated to avoid any copy right infringement.

Version 0:

Reviewer comments:

Reviewer #1

(Remarks to the Author)

Flaviviruses are divided into mosquito-borne (MBFV) and tick-borne flaviviruses (TBFV). However, most efforts are devoted on MBFV researches. The infectivity of MBFV is closely linked to the virus's maturation level. The infectivity of immature MBFV is severely abolished, although that of DENV could be partially rescued by interacting with ADE antibodies. Here, Jiri et al found that immature prM-containing particles of TBFV exhibited infectivity in mammalian cells under physiological conditions. In the mice model prM-TBEV still present high pathology, contrasting with the reduced pathogenicity of prM-WNV. Using alpha-fold, they constructed the structures of prM-TBEV and prM-USUV and identified key residues involved in the pH-dependence of prM/E interactions. They further tested the function by mutagenesis and demonstrate these mutations render TBEV more MBFV-like. The authors found a very interesting phenomenon. However, without comprehensive mechanism study, the current results still could not support their conclusions.

Suggestions :

1. In Line 117 and Fig. 1d, they demonstrate "These results showed that prM-TBEV were generally infectious, without cell-line specificity." They should compare the infectivity of prM-TBEV and M-TBEV at the same time.
2. In the figure 4, they modeled the immature structure of TBEV and USUV with alpha-fold with available sequences and structures from the database. However, none of the reference structures used includes immature prM (Ref. 18, 37, 62, 66.). The prM structure is highly flexible, without crystal or cryo-EM structure resolution, the structure data here is not convincing.
3. In figure 4, they chose USUV as the representative of MBFV. However, there are very limited structural studies on USUV. Why not use DENV or ZIKV, which have very high-resolution structures of immature virus.
4. In figure 5, they introduced mutations in the prM-E interface and found interactions within the prM-E complex influenced pH sensitivity, furin cleavage site accessibility, and virus infectivity. However, they need to validate whether these mutations really affect the pH-dependent binding between prM and E protein. Otherwise, they couldn't build the connection between prM-E interaction and TBEV maturation.
5. Line 117, they stated "without cell-line specificity". Actually, there is a big variation of infectivity among cell lines in Figure 1D.
6. In line 86, the description of "prM conformation changes" is not accurate. It's prM-E or envelope protein.
7. In Fig. 1b, the molecular weight of the E protein of mature and immature viruses should be the same. Why there is an obvious difference?

Reviewer #2

(Remarks to the Author)

Holoubek et al report a very interesting study showing that immature particles of tick-borne flaviviruses are fully infectious, in stark contrast to the better studied mosquito-borne flaviviruses, which are not. The study is very carefully done, and the results are convincing.

I disagree, however, on their interpretation when they indicate that the smooth form of the virus, formed of 90 E dimers, cannot form unless prM is cleaved by furin, and that the virus would remain in a sort of intermediate state, which is difficult to understand based on structural considerations. The available structural results on the flavivirus envelope proteins indicate that the interactions of the residues forming the prM furin site with the E protein are broken during the immature-spiky to immature transition, allowing this segment to become a disordered loop exposed at the surface of the E dimers, where it can

be cleaved by furin. The structure of the (pr/E)₂ heterotetramer at low pH (PDB accession 7QRF), which contains the residues of the furin site, shows that they are disordered and are not making the interactions described for the immature spiky form, which the authors used to reach the conclusion that the dimer cannot form.

Structural studies with the ectodomain of the flavivirus E protein have shown that the dimer of the tick-borne viruses is much more stable than its counterpart in mosquito-borne flaviviruses. This observation can also explain the irreversibility of the spiky-to smooth transition: once the E dimers form, they do not dissociate, with the pr portion tethered via the uncleaved furin site, which can be easily cleaved by furin during entry, as the authors very nicely show here. The more labile E dimer of the mosquito-borne flaviviruses reversibly dissociates to reform the spiky form when exposed back to neutral pH when prM is uncleaved. The mutagenesis experiments on the E protein made in the present paper do not have such a clearcut effect as interpreted by the authors and could be the result of a destabilization of the E dimer, making it more similar to those of the mosquito-borne flaviviruses.

In sum, the authors report a very important finding. I only challenge their interpretation of the results. It is very important that they leave the door open to an alternative interpretation based on the experimental structural data available, as theirs is based on modeling that could be misleading.

Reviewer #3

(Remarks to the Author)

Both tick- and mosquito-borne flaviviruses are major public health problems worldwide and are expected to threaten new regions each year. It is thus critical to understand the similarities and differences between tick- and mosquito-borne flaviviruses in terms of their virology, immunology, and pathogenesis. In this manuscript, the authors compared the in vitro (cell lines) and in vivo (mice) infectivity of mature (ME containing virions) vs immature (prME containing virions) TBEV (tick-borne) and WNV (mosquito-borne). As expected, mature WNV exhibited higher infectivity in cells and pathogenesis in mice than immature WNV; in contrast, both mature and immature TBEV viral preparations showed similar levels of infectivity in cells and pathogenesis in mice, highlighting an important difference in the biology of these 2 flaviviruses. These results are significant for the field and will guide development and testing of both tick- and mosquito-borne flaviviruses antivirals and vaccines. However, as described below, the mouse data lack rigor.

Specific comments:

Figure 3: As the GE:PFU ratio for prM-WNV is high as compared with GE:PFU ratio for m-WNV, it is surprising to see that 1,000 GE of prM-WNV and 1,000 GE of M-WNV induce similarly severe disease in mice, as measured via survival rate and clinical score. The viral preparations used in this study are not recombinant viruses and may contain some m-WNV in prM-WNV stock and vice versa. In addition, this experiment was performed only once with n = 5 mice/group, with unknown male vs female numbers. In the absence of recombinant viruses, I suggest performing these mouse survival experiments with at least 3 different viral preparations, measuring both GE and PFU values (I agree with the authors that the mice should be infected based on GE values; but the PFU values should be provided in order to interpret the results). In addition to survival and clinical score, weight loss should be measured. Finally, the survival/weight loss/clinical score data should match viral load data in mice—the authors can assess level of infectious virus in few tissues at one of the key time points after infection.

Version 1:

Reviewer comments:

Reviewer #1

(Remarks to the Author)

The revised manuscript, authored by Holoubek et al, had significant improvements over the last version. I congratulate the authors on their effort. However, I think there are still major issues to be addressed.

(1) Fig.1C, How can you do statistics analysis between PFU and RNA copies? Also in 2B.

(2) In Fig.1C, the PFU titer of prM-TBEV is about 2 logs lower than that of the M-TBEV. Please clarify the cell line for this assay? PS or Vero? However, in Fig. 1D, the titers of prM-TBEV and M-TBEV on PS, Vero, BHK21, and IRE are similar. How to explain this?

(3) In figure 5, lack of experimental data to support the introduced mutations in the prM-E interface influence the pH sensitivity of furin cleavage of prM-TBEV.

(4) Why does the pH dependence of furin cleavage of prM-TBEV this MS differ from that in 'Structure of immature tick-borne encephalitis virus'? Figure S2 in this MS and Figure 4 in 'Structure of immature tick-borne encephalitis virus'

Minor points:

(1) Fig.1e and 1f, Line 123 MOI=1, Line 620 MOI=0.1, which correct?

(2) Fig.2d, why is GE:PFU<1 for mTBEV?

(3) Fig.3I, 'm-WNV', not 'm-TBEV'.

(4) Fig.1A, they color the LoVo cells with different density. It does not make sense, as this is a cell line.

(Remarks to the Author)

I have read with keen interest the revised version of this manuscript. The results reported clearly point to a fundamental difference in the maturation of mosquito-borne vs tick-borne flaviviruses. The weak point, though, is the lack of experimental structural data to explain their observations, especially that the authors do not provide the reliability metrics for the AlphaFold predictions used to identify the residues in interaction between the proteins prM and E. I have therefore used the newly released AlphaFold3 server (doi: 10.1038/s41586-024-07487-w) to make the structural predictions for a TBEV prM/E heterodimer and a (prM/E)₂ heterotetramer and verify the confidence metrics. The resulting atomic models indeed have high pLDDT scores in general (as expected, as there are many structures of E and prM available), except for the regions around the furin cleavage site and for the contacts between the two prM/E protomers forming the dimer that is the building block of the smooth immature particle, indicating that the interactions made by these residues are not predicted in a reliable fashion. Because of the low confidence metrics in these precise regions, the whole paragraph between lines 218 to 232 is poorly supported and ambiguous. As a result, the effect of the three mutations (N256A in E and K81G+E83R in prM) could potentially result from interfering with the interactions described in that paragraph, or could alter the interactions in a way that is different to what the authors postulate. What is true is that the residues mutated are in the interacting region between the two E/prM protomers of the dimeric building block of the immature herringbone smooth particle. That the effect of mutating these residues may not be due to what the authors have postulated does not diminish the value of their results, but they should be more open minded about the possible reasons they see this effect, and include further available data on flavivirus maturation, as discussed below.

The reason why the furin cleavage site (positions 85-RTRR-88 in TBEV prM) is inaccessible at neutral pH is very likely due to the insertion of the segment immediately downstream (positions 90-VLIPSH-95) into a groove formed by the two short helices αA and αB at the membrane-facing face of the E protein. This was shown initially in the mature particle of dengue virus (DOI: 10.1038/nsmb.2463) and also in the newly reported structure of immature and mature TBEV (ref. 27 in this manuscript). The inserted segment is such that the side chains of prM His95 (or His 7 in protein M numbering) packs against the side chain of His 216 of E (both histidines are strictly conserved across flaviviruses). Protonation of these histidines at mildly acidic pH leads to dissociation of the inserted segment because of repulsion of the two apposed positively charged side chains, as predicted for dengue virus in the initial paper DOI: 10.1038/nsmb.2463, and by experimentally showing that mildly acidic pH induces dissociation of the E/M heterodimer extracted from TBEV particles (see doi: 10.15252/embr.202050069, Figure 3). These interacting histidines constitute one of the important pH sensors of the mature virion for activation for fusion in the acidic endosomal environment in mature virions. The data presented in this manuscript now also indicates that release of the inserted segment from the $\alpha A/\alpha B$ groove in E also leads to exposure of the furin cleavage in immature particles.

Another important element that sheds light on the results of this paper is the fact that for TBEV, the pr moiety of prM stabilizes the E dimer at acidic pH, as shown in ref. 38 of the evaluated manuscript. That paper also showed that at neutral pH, the E0F0 loop in domain I (termed "glycan" loop, as it carries the only N-linked glycosylated asparagine in E in TBEV and in most flavivirus E), actively knocks out prM from the binding site, acting as a relay in stabilizing the E dimer at neutral pH. Those results are compatible with the non-reversibility of the conformational change of immature particles, such that, once they have formed the herringbone smooth conformation, the particles stay that way, either because the pr moiety stabilizes the E dimers at acidic pH, or because the glycan loops does it at neutral pH. One explanation of the results presented in this manuscript is that upon locking the E dimer at neutral pH, the glycan loop may also block access of the 90-VLIPSH-95 segment to insert back into the $\alpha A/\alpha B$ groove underneath the E protein, and as a result, the furin cleavage site preceding this segment stays exposed at the particle surface, in the absence of any specific interaction of the pr moiety with its binding site at the E dimer interface, since the glycan loop is closed (as explained in reference 38). The mutations that the authors have introduced at the interface may interfere with this process. Furthermore, subsequent studies have shown that stabilization by pr of the E dimer is not observed for mosquito-borne viruses (see doi: 10.1128/mbio.00706-23, a study that focused on yellow fever virus pr/E complex, but which shows that the same for two other mosquito-borne viruses, Zika virus and dengue virus, for which pr does not stabilize the E dimer at acidic pH as it does for TBEV).

The key to explain the results observed here is that the furin cleavage site in TBEV and in other tick-borne flaviviruses is exposed at neutral pH and can be accessed by furin at the particle surface. Superposing the (pr/E)₂ structure (PDB 7QRE) on the E protein of the PDB 7Z51 structure of the mature TBEV particle provided in ref. 27 shows that the distance between the last prM residue observed in 7QRE (Gly 80, 5 residues before the furin site) to Leu20 in M (which is Leu108 in prM numbering), a residue roughly at the point where the M polypeptide chain turns to allow the 90-VLIPSH-95 segment to enter the $\alpha A/\alpha B$ groove of E) is at most at a distance of 40Å, which can easily be spanned by the intervening 20 amino acids from the furin site to this point (aa 88-108).

In mosquito-borne viruses, the herringbone pattern is not present at neutral pH in immature particles, as they switch back to the spiky form as soon as the pH is raised to neutral and the 90-VLIPSH-95 segment inserts back into the $\alpha A/\alpha B$ groove, which restricts access of furin to the cleavage site, which is immediately upstream this segment. In contrast, in TBEV the herringbone is maintained for the reasons stated above, with the furin site accessible as the 90-VLIPSH-95 segment cannot snap back into its binding groove. If prM is not cleaved at neutral pH before endocytosis, exposure to low pH would lead to opening of the glycan loop to accommodate prM in the endosomes, and membrane fusion for entry would be inhibited. It is important therefore that the cleavage takes place at the cell surface, before particle uptake.

To summarize, the mutations that were tested in this work, based on the AlphaFold model, correspond to residues that can alter the interaction between prM and E in a way that is difficult to account without a clear visualization of the actual interactions they are making in the particle. They could affect the closing of the glycan loop to lock the E dimer such that the segment downstream the furin site can still enter the $\alpha A/\alpha B$ groove of E so that the site is inaccessible to furin at neutral pH for instance, explaining the results obtained. But the manuscript shows that altering residues at the E/prM interface near the furin site has a clear impact in the behavior of the TBEV particles, making them more similar to the mosquito borne viruses,

and perhaps interfering with the irreversibility of the conformational change that leads to maturation in tick-borne viruses.

In addition to these main concerns, there are minor points in the manuscript that the authors should consider:

Legend to Figures: please specify the cells used in each case the Figure legends (in line 623 for Figure 1; 632 for Figure 2, etc). Also, in the legend to Figure 2e (lines 635-636), it is not clear if the particles were exposed to acidic pH for cleavage by r-furin, and then put back at neutral pH to measure the infectivity, or whether the infection of LoVo cells was done at neutral or at acid pH, by exposing the cells to low pH. Please clarify.

Line 196: should be Figs. S6 and S7 rather than S3, S4 as stated.

Reviewer #3

(Remarks to the Author)

The authors did a great job by performing additional experiments to address my prior comments related to viral infectivity used to infect mice and measurement of viral load in target tissues (serum and brain).

They however performed all experiments with only female mice. Rationale for not using both male and female mice are not stated.

Finally, Number of independent experiments still unclear. It is important to be transparent about experimental repeats, number of animals per group in each experiment, and male vs female animals.

Version 2:

Reviewer comments:

Reviewer #1

(Remarks to the Author)

The authors made substantial change on the MS and they answered all the concerns I raised. I have no further questions.

Reviewer #5

(Remarks to the Author)

As requested, I am reviewing only the results that pertain to reviewer 2's comments. I will make four main points:

1. I differ in the view that the unzipping of the linker downstream of the FCS is an established fact. The main arguments in favor of this mechanism are summarised in lines 241-244 as (1) H216 in E and H95 in prM are directly apposed and act as a pH switch; and (2) the prM linker binding to E is disrupted by acidification of detergent extracted heterodimers.

Upon inspection of published structures, H216 is often positioned away from the groove for both the mature (e.g. PDBID 5O6A for TBEV; 7KV8, 8Y3G for DENV) and immature (PDBID 8PUV for TEBV 6ZQI; for Spondweni and Binjari, the equivalent positions are not a histidine). Protonation of H95 is not unambiguously predicted to result in the destabilisation of the linker (e.g. in 8PUV). The residue is only partially buried and at the end of the linker, which binds to E through several hydrophobic interactions. Indeed, the pH-induced disruption described above (Ref 35) is observed in presence of detergent, which would be a major co-contributor to the release (possibly mimicked by the E stem during fusion). Ref 25 also cited in the 241-244 section, mostly focuses on a H98A mutant shown to abolish low pH-induced release of prM from E. While supporting the role of H98 in this transition, it does not specifically demonstrate unzipping of the linker region.

Thus, while the conclusion that "the FCS remains exposed in smooth immature particles" remains valid, the rest of the section around lines 268-271 is speculative ("upon re-exposure to neutral pH, the stabilization of the dimer interface by the glycan loop of TBEV E is such that the zipper cannot gain access to the αA - αB groove underneath the E dimer interface"). For example, it is possible that the destabilisation of the pr-E interface at neutral pH and the inability to reassemble a spiky particle are sufficient to leave the FCS exposed without invoking differential zipping/unzipping of the downstream linker.

2. The second point relates to the mutations N256A+K81G+E83R. These mutations are hypothesised to allow rezipping in TBEV. I agree with reviewer #2 that the rationale is highly hypothetical in the absence of further structural analyses.

For instance, mutation of K81 to G not only removes the side chain charge but also alters the flexibility of the whole region preceding the FCS. It may therefore have a direct effect in facilitating the FCS exposure, which may completely differ from the "gliding" and "snap back" mechanism proposed line 287-289.

An experimental structure of the mutant may not provide much further insights since the rationale is mostly based on changes in the dynamics of this region rather than the static structure. Molecular dynamics simulations however may provide useful insights to support the rationale that the 3 mutations restrict FCS exposure.

In absence of such data, much of the corresponding result (lines 284-295) and discussion sections (lines 369-371) should be rewritten (including Fig. 6) to account for the hypothetical nature of the impact of the mutations on the structure and dynamic of prM-E, and uncertainty about the exact location of the zipper itself and the hypothesised "tighter E dimer interaction".

3. The mutations will also have an impact on the neutral-pH immature virion. Are the virions for the wild type and mutant in the state they are assumed to be? That is, smooth at neutral pH rather than the spiky structure of immature MBFVs.

4. Connectivity and topology correspond to 10.1128/JVI.00197-13 (2013) in Fig 5 and Supp Fig7 USUV. This has been shown to be incorrect in Ref 43, and confirmed by Ref 33 for TBEV and 10.1128/jvi.01809-24 for DENV VLPs. Please update M (and all TMs in Fig. 5).

Minor point: it would be interesting to see BinJV integrated in analyses of the furin sites etc. (e.g. Supp Fig. 5-7). It is the most complete structure for an immature flavivirus particle, has a somewhat similar biology to the immature TBEV analysed here, but harbours a different sequence around the FCS.

Fig 6a: it would make sense to compare to the experimental structure(s) that have the FCS/linker, at least the immature, neutral pH TBEV. Inset in Fig 6b: it should be clearly stated that this is a proposed model and not structural data.

Version 3:

Reviewer comments:

Reviewer #5

(Remarks to the Author)

The revised manuscript is much improved. The addition of the MD supports the proposed model of E dimer destabilisation/stabilisation that may regulate FCS exposure. While the absence of cryo-EM images establishing the smooth or spiky morphology of key structures remains a weakness of the paper, this could be addressed by minor changes in the discussion and figure.

Specifically, mentions of the herringbone pattern and spiky form of the particles should be toned down and presented as plausible scenarios when there are no corresponding cryo-EM images or structures (e.g. l. 394-398 “the herringbone pattern of E dimers remains...” and l. 416 “allowing the reversion to the spiky form at neutral pH”). Indeed, the spiky/smooth conformation is prominent in Fig. 6. Most readers will assume that these structures are known if not otherwise stated. A question mark or dotted line would alert the reader of models that remain hypothetical (e.g. neutral pH immature after acidification, mutant at neutral pH).

Response to the Reviewers

Reviewer 1

1. In Line 117 and Fig. 1d, they demonstrate “These results showed that prM-TBEV were generally infectious, without cell-line specificity.” They should compare the infectivity of prM-TBEV and M-TBEV at the same time.

Response: In response to the reviewer's suggestion, we have completed the infectivity comparison. Figure 1d has been updated to include data for both prM-TBEV and m-TBEV across all tested cell lines.

2. In the figure 4, they modeled the immature structure of TBEV and USUV with alpha-fold with available sequences and structures from the database. However, none of the reference structures used includes immature prM (Ref. 18, 37, 62, 66.). The prM structure is highly flexible, without crystal or cryo-EM structure resolution, the structure data here is not convincing.

Response: Thank you for this comment. To better address this point, we have modified the text (lines 88-89, 348-351, 354-359, 365-375), added additional data, and included supporting information from our recent structural study to make the information more convincing. Figures 4d and 4e illustrate the fully smooth conformation, indicating that this conformation cannot exist without the cleavage of the furin site, as the position of the prM protein is central to the tetrameric complex. Consequently, the model in Figure 4 depicts the prM in a fully smooth conformation, as seen in the PDB 7Z51 cryo-EM structure of the TBEV virion. This suggests that furin cleavage must occur before the formation of the fully smooth conformation because, in the tetrameric smooth complex, the furin cleavage site would otherwise cause steric clashes with the E protein. Our structural modelling hypothesis aligns with recently published cryo-EM data (<https://www.biorxiv.org/content/10.1101/2023.08.04.551633v1>, now in press in Science Advances – full text uploaded for review; Figures 3e and S8), which show that the M protein N-terminus location beneath the E-DII implies prM cleavage precedes formation of the herringbone structure of the mature particle. This is supported by the inaccessibility of the furin cleavage site in both the virus's surface and preassembled E ectodomain dimers at low pH. These findings corroborate our hypothesis and the results presented in our paper.

PDBs: 7Z51 cryo-EM of TBEV virion, with 7QRF (black structure is E protein, and orange structure is “pr” part of prM protein with magenta residue being the last residue of “pr” part that is shown with coordinates) aligned to the 7Z51. Between the two magenta residues, of these two PDB structures, there are 9 missing residues, **GKQEGSRTRRS**. With the help of AlphaFold we modelled the heterodimer prM-E and together with the two PDBs, 7Z51 and 7QRF, we made the full heterodimeric model. The missing 9 residues do not have much conformational space for modelling since the distance between the purple residues is just enough for the 9 residues to connect them. These 9 residues are in the middle of the full complex shown in the 7Z51, therefore they would form the clashes with the E protein in a smooth conformation, unless they are previously cleaved.

AlphaFold model

3. In figure 4, they chose USUV as the representative of MBFV. However, there are very limited structural studies on USUV. Why not use DENV or ZIKV, which have very high-resolution structures of immature virus.

Response: Thank you for the comment. To better align with our other biological experiments, we preferred to use USUV, which is also very similar to WNV used in other experiments within our manuscript. Figure 4 shows the reference structure of a fully smooth conformation, not the immature (spiky) one. We used a smooth USUV structure, PDB code 7LCH, with a 2.35 Å resolution.

4. In figure 5, they introduced mutations in the prM-E interface and found interactions within the prM-E complex influenced pH sensitivity, furin cleavage site accessibility, and virus infectivity. However, they need to validate whether these mutations really affect the pH-dependent binding between prM and E protein. Otherwise, they couldn't build the connection between prM-E interaction and TBEV maturation.

Response: We agree with the reviewer and have provided additional supporting information in the revised manuscript, referencing our recent structural study (<https://www.biorxiv.org/content/10.1101/2023.08.04.551633v1>, now in press in Science Advances – full text uploaded for review). Based on the tested mutations, their effect on pH dependence, and their position in the interface of the prM-E dimer, we propose the following hypothesis: The dissociation of the furin cleavage site from the E protein is crucial since furin cannot access the cleavage site unless this dissociation occurs first. Furin-dependent infectivity supports our hypothesis regarding the importance of prM-E affinity around the furin cleavage site. Moreover, our data supports a recently published study using cryoEM structural analysis of immature spiky TBEV virions. This study suggests the significant role of a salt bridge between residues E243 of the E protein and R88 of the prM protein (Fig. 3 and Fig. S8). Thus, mutations introduced in the E and prM proteins in this study partially disrupt stability and affect the pH sensitivity of the prM-E complex, as confirmed by our experimental data. We remain open to other

potential mechanisms involving the tested mutations and the in-depth analysis of pH dependency. We mentioned the limitation of our approach now in the Discussion (Lines 363-364).

5. Line 117, they stated “without cell-line specificity”. Actually, there is a big variation of infectivity among cell lines in Figure 1D.

Response: Thank you for pointing this out. We have updated the text in the revised version based on the new data in Fig. 1d (see also comment #1).

6. In line 86, the description of "prM conformation changes" is not accurate. It's prM-E or envelope protein.

Response: Thank you for pointing this out. We have corrected this in the revised manuscript.

7. In Fig. 1b, the molecular weight of the E protein of mature and immature viruses should be the same. Why there is an obvious difference?

Response: Thank you for pointing this out. We repeated the Western blots several times and discovered that the shift was caused by different serum concentrations used for virus production in different cell lines. After adjusting the serum concentration, the shift was no longer observed. The updated image is presented in the revised manuscript, specifically in Fig. 1b and the supplementary data.

Reviewer 2

Holoubek et al report a very interesting study showing that immature particles of tick-borne flaviviruses are fully infectious, in stark contrast to the better studied mosquito-borne flaviviruses, which are not. The study is very carefully done, and the results are convincing.

Response: Thank you very much for this positive assessment of our work!

I disagree, however, on their interpretation when they indicate that the smooth form of the virus, formed of 90 E dimers, cannot form unless prM is cleaved by furin, and that the virus would remain in a sort of intermediate state, which is difficult to understand based on structural considerations.

Response: To better address this issue, we have included additional data and referenced our new structural study in the revised manuscript. Specifically, we present the following:

PDB: 7Z51 cryo-EM of TBEV virion

This structure shows the homodimer of heterodimers – E protein (cyan) and M part of the prM protein (orange). One heterodimer is shown as a cartoon, and the other as a transparent surface. The M part starts immediately after the furin cleavage site. The magenta residue shown in both figures is the first residue after the furin cleavage site (**RTRRSVLIP**).

We used PDBs 7Z51 (cryo-EM of TBEV virion) and 7QRF (black structure is E protein, and orange structure is the "pr" part of the prM protein with the magenta residue being the last residue of the "pr" part that is shown with coordinates) aligned to 7Z51. Between the two magenta residues of these two PDB structures, there are 9 missing residues: **GKQEGSRTRRS**. Using AlphaFold, we modelled the heterodimer prM-E and, together with the two PDBs (7Z51 and 7QRF), created the full heterodimeric model. The missing 9 residues do not have much conformational space for modelling since the distance between the purple residues is just enough for the 9 residues to connect them. These 9 residues are in the middle of the full complex shown in 7Z51; therefore, they would form clashes with the E protein in a smooth conformation unless they are cleaved beforehand.

For clarity and completeness, we have added these data as a supplementary figure (Fig. S4). Our hypothesis, based on structural modelling, is supported by recently published structural data from an experimental cryo-EM study (preprint at <https://www.biorxiv.org/content/10.1101/2023.08.04.551633v1>, now in press in Science Advances – full text uploaded for review; Fig. 3e and Fig. S8) showing that:

"The location of the M protein N-terminus beneath the E-DII suggests that prM cleavage precedes the formation of the mature particle's herringbone structure, as the furin cleavage site would otherwise be inaccessible. Comparing to the low pH structure of a fragment of pr complexed with preassembled E ectodomain dimers, even there, the furin cleavage site would be inaccessible from the surface of the virus."

This evidence supports our hypothesis and are in line with the results presented in our paper.

The available structural results on the flavivirus envelope proteins indicate that the interactions of the residues forming the prM furin site with the E protein are broken during the immature-spiky to immature transition, allowing this segment to become a disordered loop exposed at the surface of the E dimers, where it can be cleaved by furin. The structure of the (pr/E)₂ heterotetramer at low pH (PDB accession 7QRF), which contains the residues of the furin site, shows that they are disordered and are not making the interactions described for the immature spiky form, which the authors used to reach the conclusion that the dimer cannot form.

Response: Thank you for this comment. For a detailed response, please, see the comment with the models above.

Structural studies with the ectodomain of the flavivirus E protein have shown that the dimer of the tick-borne viruses is much more stable than its counterpart in mosquito-borne flaviviruses. This observation can also explain the irreversibility of the spiky-to smooth transition: once the E dimers form, they do not dissociate, with the pr portion tethered via the uncleaved furin site, which can be easily cleaved by furin during entry, as the authors very nicely show here. The more labile E dimer of the mosquito-borne flaviviruses reversibly dissociates to reform the spiky form when exposed back to neutral pH when prM is uncleaved. The mutagenesis experiments on the E protein made in the present paper do not have such a clearcut effect as interpreted by the authors and could be the result of a destabilization of the E dimer, making it more similar to those of the mosquito-borne flaviviruses.

Response: Thank you for your comment. We appreciate your insights regarding the limitations of our approach. In our revised manuscript, we have bolstered our conclusions by referencing our recent structural study. Our assertion that the homodimer of heterodimers depicted in PDB 7Z51 requires prior furin cleavage specifically pertains to the full prM and E protein complex. While it remains plausible that this complex could form before furin cleavage, it would assume a different conformation than what is traditionally understood as the "smooth conformation."

This hypothesis is supported by recent cryoEM data on immature spiky TBEV virions, emphasizing the necessity of furin cleavage preceding the formation of the herringbone structure (preprint available at <https://www.biorxiv.org/content/10.1101/2023.08.04.551633v1>, now in press in Science Advances – full text uploaded for review). This additional reference aligns with our findings and we believe that also enhances the robustness of our conclusions.

Anyway, we mentioned the alternative explanation in the discussion in the revised version of the manuscript (Lines 364-372).

In sum, the authors report a very important finding. I only challenge their interpretation of the results. It is very important that they leave the door open to an alternative interpretation based on the experimental structural data available, as theirs is based on modelling that could be misleading.

Response: Thank you for highlighting the significance of our findings. We have taken your feedback seriously and endeavored to strengthen our manuscript by providing additional evidence and referencing supporting studies, but also mentioning the limitations of our approach. We believe these revisions have enhanced the robustness of our conclusions, thereby addressing concerns

and providing a clearer understanding of our research. Your input has been invaluable in improving the clarity and impact of our work.

Reviewer 3

Figure 3: As the GE:PFU ratio for prM-WNV is high as compared with GE:PFU ratio for m-WNV, it is surprising to see that 1,000 GE of prM-WNV and 1,000 GE of M-WNV induce similarly severe disease in mice, as measured via survival rate and clinical score. The viral preparations used in this study are not recombinant viruses and may contain some m-WNV in prM-WNV stock and vice versa. In addition, this experiment was performed only once with n = 5 mice/group, with unknown male vs female numbers.

Response: Thank you for this comment. We have now provided this information in the revised version of the manuscript.

In the absence of recombinant viruses, I suggest performing these mouse survival experiments with at least 3 different viral preparations, measuring both GE and PFU values (I agree with the authors that the mice should be infected based on GE values; but the PFU values should be provided in order to interpret the results).

Response: Thank you for your comment and suggestion. We have conducted additional experiments by infecting all groups of mice with 100 genomic equivalents per mouse using newly prepared prM-WNV virus stocks. We have also compared the genomic equivalents to plaque-forming unit ratios (GE), and the results are depicted in the attached graph below. Consistent with our previous findings, the prM-WNV preparations still exhibit a comparable GE ratio, despite containing residual mature particles. We believe that these results support the consistency of our observations.

In addition to survival and clinical score, weight loss should be measured.

Response: Thank you for your suggestion. We have repeated the experiments as recommended. In the revised version of our manuscript, we have included a graph depicting the percentage body weight change for mice infected with m/prM-TBEV and m/prM-WNV over a period of 14 days post-infection.

Finally, the survival/weight loss/clinical score data should match viral load data in mice—the authors can assess level of infectious virus in few tissues at one of the key time points after infection.

Response: Thank you for your suggestion. In response, we have updated Figure 3 in the revised manuscript to include the levels of infectious virus detected in serum at 3 days post-infection (DPI) and in the brain, a target organ, at 8 DPI for both viruses. These new results directly align with our previous observations regarding the differences in infectivity and pathogenesis between tick-borne flaviviruses (TBFV) and mosquito-borne flaviviruses (MBFV).

Response to the Reviewers

We thank all three reviewers for valuable comments on our manuscript. We did our best to modify the manuscript accordingly.

Reviewer #1 (Remarks to the Author):

The revised manuscript, authored by Holoubek et al, had significant improvements over the last version. I congratulate the authors on their effort. However, I think there are still major issues to be addressed.

(1) Fig.1C, How can you do statistics analysis between PFU and RNA copies? Also in 2B.

Response: The reviewer is correct; we have now removed the statistical analysis from these experiments and the corresponding figures. We show statistics only in graphs comparing the ratio.

(2) In Fig.1C, the PFU titer of prM-TBEV is about 2 logs lower than that of the M-TBEV. Please clarify the cell line for this assay? PS or Vero? However, in Fig. 1D, the titers of prM-TBEV and M-TBEV on PS, Vero, BHK21, and IRE are similar. How to explain this?

Response: Thank you for your comment on this, but this seems to be a misunderstanding. prM-TBEV in Figure 1C represents a result of the plaque assay (performed with PS cells) with a sample harvested from LoVo cells (see 1A). In 1D, all tested cell lines were infected with prM-TBEV or m-TBEV at MOI=0.1 and the medium was harvested 48 hpi. A plaque assay with PS cells was performed to determine the titers in these samples.

(3) In figure 5, lack of experimental data to support the introduced mutations in the prM-E interface influence the pH sensitivity of furin cleavage of prM-TBEV.

Response: This part was extensively modified and updated in the revised version of the manuscript, along with appropriate discussion. These additional structural data were prepared by our new collaborator and new co-author Prof. Félix A. Rey from the Institut Pasteur, and are in support of our biological experiments. The new data are presented at lines 277-324.

(4) Why does the pH dependence of furin cleavage of prM-TBEV in this MS differ from that in 'Structure of immature tick-borne encephalitis virus'? Figure S2 in this MS and Figure 4 in 'Structure of immature tick-borne encephalitis virus'

Response: The structure reported in our paper by Anastasina et al. (2024, Science Advances) and the related furin cleavage experiment represent a distinct context from the prM-TBEV experiment described in the present manuscript. In Anastasina et al., the experiment utilized "spiky" immature TBEV, which had not been exposed to acidic pH. Our work, however, uses "smooth" immature TBEV particles from furin-deficient cells. To generate "spiky" immature TBEV, acidification was inhibited with ammonium chloride to maintain a neutral pH environment, either in the endoplasmic reticulum or in the medium. In this case, the treatment with low pH has two effects: first, it induces pH-dependent structural rearrangements (see Fig. 5), and second, it influences the accessibility of the furin cleavage site (FCS), thus impacting furin cleavage. Our Western blot analysis corroborates structural predictions and enzyme assays (Figs. 4 and 5), demonstrating that furin cleavage is feasible in "smooth" immature TBEV across the pH range tested.

Minor points:

(1) Fig.1e and 1f, Line 123 MOI=1, Line 620 MOI=0.1, which correct?

Response: Corrected (MOI = 0.1).

(2) Fig.2d, why is GE:PFU<1 for mTBEV?

Response: Thank you for raising this point. In theory, the ratio should indeed approach 1. However, due to RNA extraction losses and the differing sensitivities of the methods we compared, we consistently observed a GE ratio of slightly less than 1. We believe this reflects a limitation of the RT-qPCR rather than an issue affecting the overall findings or conclusions of our study.

(3) Fig.3l, 'm-WNV', not 'm-TBEV'.

Response: Thank you, this was indeed a typo. Corrected.

(4) Fig.1A, they color the LoVo cells with different density. It does not make sense, as this is a cell line.

Response: Corrected – the colours for all cells were unified.

Reviewer #2 (Remarks to the Author):

I have read with keen interest the revised version of this manuscript. The results reported clearly point to a fundamental difference in the maturation of mosquito-borne vs tick-borne flaviviruses. The weak point, though, is the lack of experimental structural data to explain their observations, especially that the authors do not provide the reliability metrics for the AlphaFold predictions used to identify the residues in interaction between the proteins prM and E. I have therefore used the newly released AlphaFold3 server (doi: 10.1038/s41586-024-07487-w) to make the structural predictions for a TBEV prM/E heterodimer and a (prM/E)₂ heterotetramer and verify the confidence metrics. The resulting atomic models indeed have high pLDDT scores in general (as expected, as there are many structures of E and prM available), except for the regions around the furin cleavage site and for the contacts between the two prM/E protomers forming the dimer that is the building block of the smooth immature particle, indicating that the interactions made by these residues are not predicted in a reliable fashion. Because of the low confidence metrics in these precise regions, the whole paragraph between lines 218 to 232 is poorly supported and ambiguous. As a result, the effect of the three mutations (N256A in E and K81G+E83R in prM) could potentially result from interfering with the interactions described in that paragraph, our could alter the interactions in a way that is different to what the authors postulate. What is true is that the residues mutated are in the interacting region between the two E/prM protomers of the dimeric building block of the immature herringbone smooth particle. That the effect of mutating these residues may not be due to what the authors have postulated does not diminish the value of their results, but they should be more open minded about

the possible reasons they see this effect, and include further available data on flavivirus maturation, as discussed below.

The reason why the furin cleavage site (positions 85-RTRR-88 in TBEV prM) is inaccessible at neutral pH is very likely due to the insertion of the segment immediately downstream (positions 90-VLIPSH-95) into a groove formed by the two short helices αA and αB at the membrane-facing face of the E protein. This was shown initially in the mature particle of dengue virus (DOI: 10.1038/nsmb.2463) and also in the newly reported structure of immature and mature TBEV (ref. 27 in this manuscript). The inserted segment is such that the side chains of prM His95 (or His 7 in protein M numbering) packs against the side chain of His 216 of E (both histidines are strictly conserved across flaviviruses). Protonation of these histidines at mildly acidic pH leads to dissociation of the inserted segment because of repulsion of the two apposed positively charged side chains, as predicted for dengue virus in the initial paper DOI: 10.1038/nsmb.2463 , and by experimentally showing that mildly acidic pH induces dissociation of the E/M heterodimer extracted from TBEV particles (see doi: 10.15252/embr.202050069, Figure 3). These interacting histidines constitute one of the important pH sensors of the mature virion for activation for fusion in the acidic endosomal environment in mature virions. The data presented in this manuscript now also indicates that release of the inserted segment from the $\alpha A/\alpha B$ groove in E also leads to exposure of the furin cleavage in immature particles. Another important element that sheds light on the results of this paper is the fact that for TBEV, the pr moiety of prM stabilizes the E dimer at acidic pH, as shown in ref. 38 of the evaluated manuscript. That paper also showed that at neutral pH, the EOF0 loop in domain I (termed “glycan” loop, as it carries the only N-linked glycosylated asparagine in E in tBEV and in most flavivirus E), actively knocks out prM from the binding site, acting as a relay in stabilizing the E dimer at neutral pH. Those results are compatible with the non-reversibility of the conformational change of immature particles, such that, once they have formed the herringbone smooth conformation, the particles stay that way, either because the pr moiety stabilizes the E dimers at acidic pH, or because the glycan loops does it at neutral pH. One explanation of the results presented in this manuscript is that upon locking the E dimer at neutral pH, the glycan loop may also block access of the 90-VLIPSH-95 segment to insert back into the $\alpha A/\alpha B$ groove underneath the E protein, and as a result, the furin cleavage site preceding this segment stays exposed at the particle surface, in the absence of any specific interaction of the pr moiety with its binding site at the E dimer interface, since the glycan loop is closed (as explained in reference 38). The mutations that the authors have introduced at the interface may interfere with this process. Furthermore, subsequent studies have shown that stabilization by pr of the E dimer is not observed for mosquito-borne viruses (see doi: 10.1128/mbio.00706-23, a study that focused on yellow fever virus pr/E complex, but which shows that the same for two other mosquito-borne viruses, Zika virus and dengue virus, for which pr does not stabilize the E dimer at acidic pH as it does for TBEV). The key to explain the results observed here is that the furin cleavage site in TBEV and in other tick-borne flaviviruses is exposed at neutral pH and can be accessed by furin at the particle surface. Superposing the (pr/E)₂ structure (PDB 7QRE) on the E protein of the PDB 7Z51 structure of the mature TBEV particle provided in ref. 27 shows that the distance between the last prM residue observed in 7QRE (Gly 80, 5 residues before the furin site) to Leu20 in M (which is Leu108 in prM numbering), a residue roughly at the point where the M polypeptide chain turns to allow the 90-VLIPSH-95 segment to enter the $\alpha A/\alpha B$ groove of E) is at most at a distance of 40Å , which can easily be spanned by the intervening 20 amino acids from the furin site to this point (aa 88-108). In mosquito-borne viruses, the herringbone pattern is not present at neutral pH in immature particles, as they switch back to the spiky form as soon as the pH is raised to neutral and the 90-VLIPSH-95 segment inserts back into the $\alpha A/\alpha B$ groove, which restricts access of furin to the cleavage site, which

is immediately upstream this segment. In contrast, in TBEV the herringbone is maintained for the reasons stated above, with the furin site accessible as the 90-VLIPSH-95 segment cannot snap back into its binding groove. If prM is not cleaved at neutral pH before endocytosis, exposure to low pH would lead to opening of the glycan loop to accommodate prM in the endosomes, and membrane fusion for entry would be inhibited. It is important therefore that the cleavage takes place at the cell surface, before particle uptake.

To summarize, the mutations that were tested in this work, based on the Alphafold model, correspond to residues that can alter the interaction between prM and E in a way that is difficult to account without a clear visualization of the actual interactions they are making in the particle. They could affect the closing of the glycan loop to lock the E dimer such that the segment downstream the furin site can still enter the $\alpha A/\alpha B$ groove of E so that the site is inaccessible to furin at neutral pH for instance, explaining the results obtained. But the manuscript shows that altering residues at the E/prM interface near the furin site has a clear impact in the behavior of the TBEV particles, making them more similar to the mosquito borne viruses, and perhaps interfering with the irreversibility of the conformational change that leads to maturation in tick-borne viruses.

Response: Thank you very much for this insightful comment. We fully agree with the reviewer's interpretation. With the assistance of Prof. Félix A. Rey, who has joined as a co-author, we have added extensive new structural data that align well with the interpretations suggested by the reviewer. These additions are now presented in lines 229–324 and illustrated in Figures 4d–f, 5a, and 6a–c.

More specifically, using the AlphaFold3 server we generated predictions for the (prM-E)₂ dimer that align closely with the known structure of the TBEV (pr/E)₂ dimer (PDB: 7QRE). This structure shows pr density up to residue Gly80, which follows the stabilizing disulfide bond (Cys43-Cys80) in the pr domain. In the AlphaFold3 predictions, the downstream residues of full-length prM zipper along the E dimer at neutral pH, burying the furin cleavage site (FCS) within the E dimer interface (Fig. 4f, Supplementary Fig. 7). The predicted conformations of C-terminal residues suggest they are disordered, consistent with cryo-EM data for the virion at neutral pH. These findings indicate that the zipper's alignment within the $\alpha A-\alpha B$ groove of E conceals the FCS from furin cleavage under neutral conditions.

Our new data show that immature TBEV virions are cleaved by furin on cell surfaces, while immature MBFVs are not (Fig. 5a, b). Upon neutral pH re-exposure, the glycan loop of TBEV stabilizes the E dimer, preventing the zipper from reaching the $\alpha A-\alpha B$ groove and keeping the FCS exposed. This allows furin to cleave prM, shedding pr before endosomal uptake, where low pH reopens the glycan loop to accommodate the pr moiety and block fusion. In MBFVs, prM likely re-zippers into the E groove at neutral pH, exposing the FCS only at endosomal acidic pH, thereby inhibiting fusion (Fig. 5b).

To test the hypothesis that E dimer stabilization is key to FCS exposure at neutral pH, we mutated selected residues in the predicted prM-E complex. We focused on interactions unique to MBFV complexes that might allow the zipper to close the E dimer interface and block FCS. We noted that TBEV prM residue Lys80 forms a hydrogen bond to the pr domain in all top-ranked predictions (Fig. 6a). Disrupting this bond might increase linker flexibility, promoting zipper re-entry into the binding groove along the E dimer. Additionally, we observed E protein residue Asn256, which stabilizes two β -strands at the dimer interface, rigidifying this region (Fig. 6a). Altering this interaction might enhance prM re-zipping. Finally, we noted that prM residue Glu82, which faces the opposite E dimer, is positively charged in MBFVs (Supplementary Fig. 5), suggesting further stabilization differences. These

structural data were in agreement with the results we obtained from biological experiments with the authentic viruses.

In addition to these main concerns, there are minor points in the manuscript that the authors should consider:

Legend to Figures: please specify the cells used in each case the Figure legends (in line 623 for Figure 1; 632 for Figure 2, etc).

Response: Corrected.

Also, in the legend to Figure 2e (lines 635-636), it is not clear if the particles were exposed to acidic pH for cleavage by r-furin, and then put back at neutral pH to measure the infectivity, or whether the infection of LoVo cells was done at neutral or at acid pH, by exposing the cells to low pH. Please clarify.

Response: The particles were exposed to acidic pH for cleavage by r-furin, and then put back at neutral pH to measure the infectivity. Corrected.

Line 196: should be Figs. S6 and S7 rather than S3, S4 as stated.

Response: Corrected.

Reviewer #3 (Remarks to the Author):

The authors did a great job by performing additional experiments to address my prior comments related to viral infectivity used to infect mice and measurement of viral load in target tissues (serum and brain).

They however performed all experiments with only female mice. Rationale for not using both male and female mice are not stated.

Response: The experiments were conducted exclusively with female mice, in adherence to our ethical approval. Female mice are commonly used as a valuable model in TBE research, and the rationale for this choice has now been clarified in the manuscript (Line 531), along with examples of references to other recent studies that also employ only female mice for TBEV experiments (references 67-72).

Finally, Number of independent experiments still unclear. It is important to be transparent about experimental repeats, number of animals per group in each experiment, and male vs female animals.

Response: We have now added all details on the number of independent experiments and numbers of animals used in each experiment, both in methods and in figure legends.

Response to the reviewer

We thank Reviewer 5 for the insightful and constructive comments, which significantly improved the clarity and interpretation of our manuscript. Your observations helped us recognize that our previous presentation of the “zippering/unzippering” concept regarding the prM linker may have been misleading. In the revised manuscript, we now clarify that, based on available experimental structures at neutral pH, the scissile bond within the prM linker is retracted due to its interactions with protein E, rendering it inaccessible to furin. Acidic pH exposure must therefore cause this linker segment to detach from E, thereby allowing furin cleavage.

Following your suggestion, we have conducted molecular dynamics (MD) simulations on the predicted structure of the (prM/E)₂ dimer, comparing wild-type and mutant forms. These simulations suggest a significantly more dynamic dimer interface in the mutant, which is now illustrated in the new **Figure 5**. Although the individual effects of the two prM mutations are difficult to isolate, the mutation in protein E (N256A) appears to have a clear destabilizing effect on the E dimer. AlphaFold predictions of the sE dimer structure for the wild-type match closely with the experimental structure, as expected. However, the N256A mutant prediction shows disruption of two key dimer interactions near N256: (1) an interprotomer main-chain hydrogen bond and (2) an aromatic–proline interaction. These results suggest a weakened dimer interface.

Microsecond-scale MD simulations of both wild-type and mutant sE dimers show that these interactions are preserved in the wild type but disrupted in the mutant. This is also reflected in an increased solvent accessibility of this region in the mutant, as depicted in **Figure 5**.

Additionally, our sequence analysis across flavivirus E proteins revealed that these two stabilizing interactions are uniquely facilitated by an insertion in the fg loop of tick-borne flaviviruses (TBFVs). This insertion contains a proline (P210), which interacts with an aromatic residue (Y255 or F255) in the j strand of the opposing E monomer. This aromatic–proline pairing (e.g., 255-YN-256 or 255-FN-256) is consistently observed in TBFVs but absent in non-TBFVs. This supports our hypothesis that these unique structural features contribute to the irreversibility of the smooth-to-spiky conformational transition observed in TBFVs.

In response to your comment:

“For example, it is possible that the destabilisation of the pr-E interface at neutral pH and the inability to reassemble a spiky particle are sufficient to leave the FCS exposed without invoking differential zipping/unzipping of the downstream linker.”

We now propose that the extra stability of the wild-type TBEV E dimer—mediated by the aforementioned interactions—prevents reformation of the spiky conformation at neutral pH, thereby maintaining the scissile site concealed. This is consistent with the original observations by Stadler et al. (1997, DOI: 10.1128/jvi.71.11.8475-8481.1997). Once cleaved, however, the linker can “zipper” back in the context of the herringbone arrangement of E dimers, as observed in mature virions, but not when uncleaved—thus leaving the furin cleavage site (FCS) exposed.

In summary, your comments were instrumental in refining our interpretation. The manuscript has been substantially revised to reflect these new insights. Unfortunately, due to current resource limitations, we are unable to repeat the experiments with the single N256A mutant in E, although we anticipate that it would show similar behavior to the triple mutant.

We now provide a point-by-point response to each of your comments (responses are marked in a different font color below).

Reviewer 5:

As requested, I am reviewing only the results that pertain to reviewer 2’s comments. I will make four main points:

1. I differ in the view that the unzipping of the linker downstream of the FCS is an established fact. The main arguments in favor of this mechanism are summarised in lines 241-244 as (1) H216 in E and H95 in prM are directly apposed and act as a pH switch; and (2) the prM linker binding to E is disrupted by acidification of detergent extracted heterodimers.

Upon inspection of published structures, H216 is often positioned away from the groove for both the mature (e.g. PDBID 5O6A for TBEV; 7KV8, 8Y3G for DENV) and immature (PDBID 8PUV for TEBV 6ZQI; for Spondweni and Binjari, the equivalent positions are not a histidine). Protonation of H95 is not unambiguously predicted to result in the destabilisation of the linker (e.g. in 8PUV). The residue is only partially buried and at the end of the linker, which binds to E through several hydrophobic interactions. Indeed, the pH-induced disruption described above (Ref 35) is observed in presence of detergent, which would be a major co-contributor to the release (possibly mimicked by the E stem during fusion). Ref 25 also cited in the 241-244 section, mostly focuses on a H98A mutant shown to abolish low pH-induced release of prM from E. While supporting the role of H98 in this transition, it does not specifically demonstrate unzipping of the linker region.

Response: We thank the reviewer for raising this important point. We acknowledge that in the previous version of the manuscript, we did not address this issue in sufficient depth.

Although prM His95 (or M His7 in the mature virus) is not consistently in direct contact with its proposed partner in all available structures, its side chain inserts into a cavity that becomes increasingly basic at pH 5, as illustrated in the **Figure R1**. This is consistent with the idea that protonation-driven changes in the local environment contribute to "unzipping" of the prM linker.

In support of this interpretation, we refer to reference 35 of the original manuscript (DOI: 10.1038/ncomms4877), where the Kielian lab demonstrated that mutation of His95 to alanine generated a virus that remained immature and required a non-physiologically low pH for prM cleavage by furin. This aligns well with our hypothesis that detachment of the linker ("unzipping") is necessary for maturation. Importantly, that study also identified second-site mutations that restored maturation at physiological pH, suggesting that other regions of the protein can also influence the conformational dynamics required for prM processing. This indicates that electrostatic repulsion between His95 and E His216 is not the sole mechanism driving unlinking of the prM linker.

To further illustrate the variability in this mechanism across flaviviruses, we note that Binjari virus lacks a histidine at the equivalent E position and circulates predominantly as immature particles (DOI: 10.1126/sciadv.abe4507). Similarly, yellow fever virus also does not encode a histidine at this site and shows distinct maturation behavior, with cleaved pr maintaining high affinity for E domain II even at neutral pH—unlike what is observed in other flaviviruses (DOI: 10.1128/mbio.00706-23).

In light of these findings, we have revised the manuscript text to clarify that the proposed "unzipping" is not solely due to electrostatic repulsion between His216 and His95. Rather, we present it as one of several contributing factors to prM detachment. We have also added **Supplementary Figure 6**, which shows that residue 216 is not a universally conserved histidine across flaviviruses.

Figure R1. Environment of M protein His 7 in the structures of mature flavivirus particles, using the coordinates indicated above. These structures were determined at neutral pH, and the electrostatic surface of protein E was calculated at the two pH values using the PDB2PQR server (DOI: [10.1093/nar/gkm276](https://doi.org/10.1093/nar/gkm276)). The environment of His7 is highly sensitive to the pH, and becomes basic at pH 5, thereby repelling a protonated side chain. The His 216 in E (TBEV numbering), which is expected to become protonated at pH 5, is not conserved in all flaviviruses (Spondweni virus, for example see Figure R2, does not have a histidine at this position). In this case, the environment of His7 does not change drastically, and unzipping is likely to depend on other interactions pH dependent interactions, as discussed in the text.

Thus, while the conclusion that “the FCS remains exposed in smooth immature particles” remains valid, the rest of the section around lines 268-271 is speculative (“upon re-exposure to neutral pH, the stabilization of the dimer interface by the glycan loop of TBEV E is such that the zipper cannot gain access to the α A- α B groove underneath the E dimer interface”). For example, it is possible that the destabilisation of the pr-E interface at neutral pH and the inability to reassemble a spiky particle are sufficient to leave the FCS exposed without invoking differential zipping/unzipping of the downstream linker.

Response: We thank the reviewer for the valuable insight and agree that the inability to reassemble a spiky particle is sufficient to leave the furin cleavage site (FCS) exposed. Our molecular dynamics simulations support this conclusion, indicating that dissociation of the E dimers—responsible for the smooth herringbone surface architecture at acidic pH—is necessary to enable re-zipping of the linker.

While there may still be some debate as to whether protonation of histidine residues (as discussed above) is the primary trigger for unlinking, the available structural data clearly show that the scissile bond is inaccessible to furin in the immature conformation. As illustrated in the **Figures R2 and R3**, the polypeptide segment containing the cleavage site cannot physically reach the furin active site in its observed position.

Figure R2 demonstrates that, for efficient cleavage, the substrate polypeptide must be flexibly connected to the rest of the protein at both its N- and C-termini—a requirement not met in the immature conformation of prM. **Figure R3** further shows that the α A- α B groove of the E protein sequesters the FCS, preventing its access to furin. The scissile bond itself is marked with a yellow arrow in the figure for clarity. We have incorporated this structural insight into **Figure 4d** of the revised manuscript. The relevant section of the text has also been substantially rewritten to incorporate these findings and to clarify the conclusions drawn from the molecular dynamics simulations, as further detailed in our response to Reviewer Comment 2.

[redacted]

Figure R3. The FCS in the structures of the prM/E complex forming the spikes of immature particles. The electrostatic surface representation covers the pr/E complex, including the globular domain of pr highlighted in green in the top-left panel. Globular pr ends at Cys 79 (TEBV numbering), highlighted with a green number 2 under the alignment of Figure R5) as it makes disulfide bond of pr with Cys 41). The downstream segment of prM, including the FCS, is shown as sticks color-coded according to atom type but with carbon atoms colored according to B factors (blue, low B factor, red, high B factor, going through green and yellow). The region shown in sticks is framed in Figure R5, with the side chains of the FCS labeled according to the nomenclature of Figure R2. Only the Binjari virus structure displayed residues of the linker 2 region of Fig. R5, which form an α -helix in their N-terminal part. A yellow arrow in each panel marks the scissile bond (labeled only in the top-left panel), which also marks the beginning of the zipper, which must unzip accommodate the FCS into the furin active site. For yellow fever virus there is no structure of immature virus, so an AlphaFold prediction was used, displayed in sticks as above, but with carbon atoms colored according to the confidence level (pLDDT scores), as in the color key below.

2. The second point relates to the mutations N256A+K81G+E83R. These mutations are hypothesised to allow rezipping in TBEV. I agree with reviewer #2 that the rationale is highly hypothetical in the absence of further structural analyses.

For instance, mutation of K81 to G not only removes the side chain charge but also alters the flexibility of the whole region preceding the FCS. It may therefore have a direct effect in facilitating the FCS exposure, which may completely differ from the “gliding” and “snap back” mechanism proposed line 287-289. An experimental structure of the mutant may not provide much further insights since the rationale is mostly based on changes in the dynamics of this region rather than the static structure. Molecular dynamics simulations however may provide useful insights to support the rationale that the 3 mutations restrict FCS exposure.

In absence of such data, much of the corresponding result (lines 284-295) and discussion sections (lines 369-371) should be rewritten (including Fig. 6) to account for the hypothetical nature of the impact of the mutations on the structure and dynamic of prM-E, and uncertainty about the exact location of the zipper itself and the hypothesised “tighter E dimer interaction”.

Response: We thank the reviewer for the suggestion. In response, we performed molecular dynamics (MD) simulations based on the AlphaFold-predicted structure of the TBEV (prM/E)₂ dimer. The three mutations—N256A in E, and K80G + E82R in prM—were introduced into the model, and 1 μs-long all-atom, explicit solvent simulations were conducted for both the wild-type ((prM-E)₂^{WT}) and mutant ((prM-E)₂^{MUT}) complexes. The systems were embedded in a heterogeneous lipid bilayer mimicking the composition of the mammalian ER membrane (Fig. 5h).

The resulting Root Mean Square Fluctuation (RMSF) profiles indicate that, overall, the three mutations exert a modest destabilizing effect on the E dimer (Fig. 5i). Nevertheless, both systems remained structurally stable over the microsecond timescale, as reflected by the median RMSF values computed for the entire (prM-E)₂ dimer (Fig. 5h-l).

Notably, the simulations revealed that the distance between the fg loop and the adjacent E protomer (specifically at the j strand interface) was significantly increased in the mutant compared to the wild type. To further probe the behavior of the fg loop, we employed the 1.9 Å resolution experimental structure of the TBEV sE dimer (PDB ID: 1SVB) and compared it with the mutant N256A using MD simulations (Fig. 5m-n). The results show that in the mutant, the inter-protomer hydrogen bond fails to persist throughout the simulation (Fig. 5m), and that the contact surface between the fg loop and the j strand becomes solvent-accessible (Fig. 5n).

These findings suggest that the N256A mutation in E is principally responsible for the observed phenotype. It disrupts key contacts between the fg loop and the j strand that appear essential for maintaining E dimer stability at neutral pH. Loss of these interactions likely facilitates dimer dissociation and prevents re-formation of the spiky immature particle, thereby maintaining the FCS in an exposed state.

3. The mutations will also have an impact on the neutral-pH immature virion. Are the virions for the wild type and mutant in the state they are assumed to be? That is, smooth at neutral pH rather than the spiky structure of immature MBFVs.

Response: We agree that it would be nice to verify this, but unfortunately the virus yields from LoVo cells are too low to visualize them by electron microscopy. Our results linking stability of the E dimer with FCS exposure suggest that immature TBFV virions exposed to low pH maintain the herringbone pattern of E dimers after being brought back to neutral pH, with the FCS exposed.

4. Connectivity and topology correspond to 10.1128/JVI.00197-13 (2013) in Fig 5 and Supp Fig7 USUV. This has been shown to be incorrect in Ref 43, and confirmed by Ref 33 for TBEV and 10.1128/jvi.01809-24 for DENV VLPs. Please update M (and all TMs in Fig. 5).

Response: We apologize for this mistake. The connectivity has been revised.

Minor point: it would be interesting to see BinJV integrated in analyses of the furin sites etc. (e.g. Supp Fig. 5-7). It is the most complete structure for an immature flavivirus particle, has a somewhat similar biology to the immature TBEV analysed here, but harbours a different sequence around the FCS.

Response: We have included BinJV in our analysis, it does not contain the TBEV specific motives (see suppl. Fig. 6A). It is not known under which conditions BINJ is cleaved by furin.

Fig 6a: it would make sense to compare to the experimental structure(s) that have the FCS/linker, at least the immature, neutral pH TBEV. Inset in Fig 6b: it should be clearly stated that this is a proposed model and not structural data.

Response: Done

Reviewer #5 (Remarks to the Author):

The revised manuscript is much improved. The addition of the MD supports the proposed model of E dimer destabilisation/stabilisation that may regulate FCS exposure. While the absence of cryo-EM images establishing the smooth or spiky morphology of key structures remains a weakness of the paper, this could be addressed by minor changes in the discussion and figure.

Response: Thank you very much, the final corrections were done accordingly.

Specifically, mentions of the herringbone pattern and spiky form of the particles should be toned down and presented as plausible scenarios when there are no corresponding cryo-EM images or structures (e.g. l. 394-398 “the herringbone pattern of E dimers remains...” and l. 416 “allowing the reversion to the spiky form at neutral pH”). Indeed, the spiky/smooth conformation is prominent in Fig. 6. Most readers will assume that these structures are known if not otherwise stated. A question mark or dotted line would alert the reader of models that remain hypothetical (e.g. neutral pH immature after acidification, mutant at neutral pH).

Response: Thank you. We agree with the reviewer that this is a limitation of the study. We have modified the text accordingly—for example, by adding the statement “...although this remains to be addressed and confirmed by direct structural data, for example from cryo-EM. The absence of cryo-EM data on these structures represents one of the limitations of the present study.” (lines 413–415), and by adding the word “likely” (line 431). In Figure 5, we used dashed lines to indicate hypothetical models, and we have also updated the figure legend to reflect this.